

# On the potential use of highly oxygenated organic molecules (HOM) as indicators for ozone formation sensitivity

Jiangyi Zhang[1], Jian Zhao[1], Yuanyuan Luo[1], Valter Mickwitz[1], Douglas Worsnop[1,2], and Mikael Ehn[1]

[1]Institute for Atmospheric and Earth System Research/Physics, Faculty of Science, University of Helsinki, Helsinki, 00014, Finland
[2]Aerodyne Research Inc., Billerica, Massachusetts, 01821, United States

**Correspondence:** Jian Zhao (jian.zhao@helsinki.fi) and Jiangyi Zhang (jiangyi.zhang@helsinki.fi)

**Abstract.** Ozone ($O_3$), an important and ubiquitous trace gas, protects lives from harm of solar ultraviolet (UV) radiation in the stratosphere but is toxic to living organisms in the troposphere. Additionally, tropospheric $O_3$ is a key oxidant, and source of other oxidants (e.g, OH and $NO_3$ radicals) for various volatile organic compounds (VOC). Recently, highly oxygenated organic molecules (HOM) were identified as a new compound group formed from oxidation of many VOC, making up a significant

source of secondary organic aerosol (SOA). The pathways forming HOM from VOC involve autoxidation of peroxy radicals ($RO_2$), formed ubiquitously in many VOC oxidation reactions. The main sink for $RO_2$ is bimolecular reactions with other radicals, $HO_2$, NO or other $RO_2$, and this largely determines the structure of the end products. Organic nitrates form solely from $RO_2$ + NO reactions while accretion products ("dimers") solely from $RO_2$ + $RO_2$ reactions. The $RO_2$ + NO reaction also converts NO into $NO_2$, making it a net source for $O_3$ through $NO_2$ photolysis.

There is a highly nonlinear relationship between $O_3$, $NO_x$, and VOC. Understanding the $O_3$ formation sensitivity to changes in VOC and $NO_x$ is crucial for making optimal mitigation policies to control $O_3$ concentrations. However, determining the specific $O_3$ formation regimes (either VOC-limited or $NO_x$-limited) remains challenging in diverse environmental conditions. In this work we assessed whether HOM measurements can function as a real-time indicator for the $O_3$ formation sensitivity based on the hypothesis that HOM compositions can describe the relative importance of NO as a terminator for $RO_2$. Given the

fast formation and short lifetimes of the low-volatile HOM (timescale of minutes), they describe the instantaneous chemical regime of the atmosphere. In this work, we conducted a series of monoterpene oxidation experiments in our chamber while varying the concentrations of $NO_x$ and VOC under different $NO_2$ photolysis rates. We also measured the relative concentrations of HOM of different types (dimers, nitrate-containing monomers, and non-nitrate monomers) and used ratios between these to estimate the $O_3$ formation sensitivity. We find that for this simple system, the $O_3$ sensitivity could be described very well

based on the HOM measurements. Future work will focus on determining to what extent this approach can be applied in more complex atmospheric environments. Ambient measurements of HOM have become increasingly common during the last decade, and therefore we expect that there already are a large amount groups with available data for testing this approach.





## 1 Introduction

Ozone ($O_3$), as a key trace gas in the atmosphere, directly and indirectly affects human lives, and it plays diametrically opposed roles in the troposphere ("bad" ozone) and stratosphere ("good" ozone) (Sandermann, 1996; Staehelin et al., 2001; Seviour, 2022). The formation and depletion of $O_3$ has been investigated over the past decades (Chapman, 1930; Crutzen, 1970, 1971; Stolarski and Cicerone, 1974; Tiao et al., 1975; Dodge, 1977). Atmospheric $O_3$ is almost entirely produced through the reaction between atomic oxygen ($O^3P$) and molecular oxygen ($O_2$) (Wang et al., 2017). In the stratosphere, the $O^3P$ source is $O_2$ photolysis with ultraviolet (UV) wavelengths below 240 nm (Chapman, 1930). Stratospheric $O_3$, which constitutes approximately 90% of Earth's atmospheric $O_3$, plays a crucial role in absorbing UV radiation in the UVB band (280-315 nm), protecting organisms on the ground from the harm of UV radiation (Gruijl and Leun, 2000; Seinfeld and Pandis, 2016). Although certain man-made substances, such as chlorofluorocarbons, were found to be responsible for significant depletion of stratospheric $O_3$, the implementation of the 1987 Montreal Protocol and its subsequent amendments has contributed to the recovery of the stratospheric $O_3$ layer (Seinfeld and Pandis, 2016; Chipperfield et al., 2017).

In the troposphere, in addition to its role as a greenhouse gas (Ehhalt et al., 2001), $O_3$ serves as a secondary air pollutant due to its detrimental impacts and indirect emissions (Seinfeld and Pandis, 2016; Nuvolone et al., 2018). Not only is it toxic, but it also participates in chemical reactions that lead to the formation of other harmful molecules (Nuvolone et al., 2018). In contrast to stratospheric $O_3$, in the troposphere, the source of $O^3P$ is $NO_2$ photolysis at wavelengths less than 420 nm (Madronich et al., 1983). However, the net formation of tropospheric $O_3$ occurs through chemical reactions involving nitrogen oxides ($NO_x=NO+NO_2$) and various volatile organic compounds (VOC) in the presence of UV light (Lelieveld and Dentener, 2000). In an ideal "clean" system without any VOC, once $O_3$ is formed, it readily converts NO back to $NO_2$ by reacting with NO, resulting in a null cycle as shown below (also illustrated as "$NO_x$ cycle" in Fig A1):

$$NO_2 + hv \xrightarrow{O_2} NO + O_3 \tag{R1}$$

$$O_3 + NO \rightarrow NO_2 + O_2 \tag{R2}$$

When VOC are present, they will be oxidized to form peroxy radicals ($RO_2$) by atmospheric oxidants, such as $O_3$ and OH (Atkinson and Arey, 2003):

$$VOC + oxidant \xrightarrow{O_2} RO_2 \tag{R3}$$

The $RO_2$ radicals can thus replace $O_3$ in converting NO into $NO_2$ (R4a). Some fraction of $RO_2$ + NO reactions will also lead to the formation of organic nitrates, $RONO_2$ (R4b) (Atkinson and Arey, 2003):

$$RO_2 + NO \rightarrow RO + NO_2 \tag{R4a}$$

$$\rightarrow RONO_2 \tag{R4b}$$

In summary, the presence of $RO_2$ radicals, supplied by VOC, can perturb the "$NO_x$ cycle", leading to the net increase of $O_3$ (Fig. A1). As one of the most characteristic features of the photochemical smog episodes in many cities (Tiao et al., 1975; Tang





et al., 1995; Dickerson et al., 1997), this $O_3$ formation process (R1-R4a, Fig A1), known as $O_3$-$NO_x$-VOC sensitivity or $O_3$
formation sensitivity, has been investigated since the last century (Haagen-Smit et al., 1953; Kinosian, 1982). The Empirical
Kinetic Modeling Approach (EKMA) curve, namely $O_3$ isopleths, was proposed by Dodge (1977) and has been widely used to
visually study the $O_3$ formation sensitivity (Liu and Shi, 2021). The $O_3$ isopleths reveal a highly nonlinear response of $O_3$ to its
precursors, demonstrating that the impact on $O_3$ formation by reducing or increasing either VOC or $NO_x$ does not consistently
exhibit the same behavior (Meyer Jr et al., 1977; Harris et al., 1982; Sillman et al., 1990). The outcome is dependent on the
relative concentrations of VOC (as a surrogate for $RO_2$) and $NO_x$, leading to the division of the formation area into $NO_x$-limited
and VOC-limited regimes (Sillman, 1999; Melkonyan and Kuttler, 2012). In the $NO_x$-limited regime, the concentration of $O_3$
generally increases with an increase in $NO_x$, while its response to changes in VOC remains relatively small. This is because the
supply of $RO_2$ species from VOC is abundant and R4a is limited by NO (the "$NO_x$ cycle" is saturated in Fig. A1). Conversely,
in the VOC-limited regime, an increase in VOC concentration generally leads to an increase in $O_3$, whereas an increase in
$NO_x$ may even result in a decrease in $O_3$ levels. This is because $NO_x$ is in excess compared to $RO_2$. Moreover, very high
levels of $NO_x$ can directly titrate $O_3$ (Sillman, 1999) or consume OH radicals (Atkinson et al., 2004), thereby reducing the
supply of $RO_2$ species (R3) and promoting the reaction R2, which results in a decrease in the $O_3$ concentration. These effects
were demonstrated by the amplified $O_3$ pollution in cities during the COVID-19 lockdown, when $NO_x$ emissions dropped
dramatically (Sicard et al., 2020).

To mitigate the uncertainties associated with photochemical models and efficiently determine the $O_3$ formation sensitivity,
various photochemical indicators have been utilized since the last century. (Wang et al., 2017; Liu and Shi, 2021). For example,
the $O_3$ production efficiency (OPE = $\frac{\Delta O_3}{\Delta NO_z}$) is defined as the number of $O_3$ molecules produced per molecule of $NO_x$ before
the $NO_x$ is oxidized to more stable products, i.e., $NO_z$ species (Trainer et al., 1993; Wang et al., 2017). Smaller values of OPE
indicate the inefficiency of the "$NO_x$ cycle" (Fig. A1), suggesting that the supply of $RO_2$ from VOC becomes the limiting factor.
As a result, the photochemical system tends to be VOC-limited. Conversely, when OPE values are higher, the system tends to
be $NO_x$-limited. Additionally, several modifications had been made to the OPE indicator to account for different situations,
such as replacing $\Delta O_3$ with $\Delta O_x$ ($O_x$=$O_3$+$NO_2$) (Kleinman et al., 2002) and replacing $\Delta NO_z$ with $\Delta NO_y$ ($NO_y$=$NO_z$+$NO_x$)
(Wang et al., 2006). Other chemical species have also been utilized as photochemical indicators, including the ratio of $H_2O_2$
to $HNO_3$ (Hammer et al., 2002). High values of $\frac{H_2O_2}{HNO_3}$ indicate high potential for cross-reactions of two $HO_2$ radicals, which
is associated with high $\frac{VOC}{NO_x}$ and thus indicative of the $NO_x$-limited regime. For a more widespread application, space-based
$\frac{HCHO}{NO_2}$ measurements from global $O_3$ monitoring satellites have been introduced as an indicator (Martin et al., 2004), based on
the fact that levels of HCHO and $NO_2$ in the tropospheric column are closely linked to VOC and $NO_x$ emissions, respectively.
However, all these indicators are not inherently linked to the $O_3$ formation process, and the corresponding threshold values
depend on environmental conditions (Liu and Shi, 2021). This makes it challenging to universally apply these indicators.

During the past decade, highly oxygenated organic molecules (HOM) have been recognized as a new group of VOC oxi-
dation products, particularly important for the formation of secondary organic aerosol (SOA) due to their fast formation and
low volatilities (Ehn et al., 2014; Bianchi et al., 2019). Aerosols play a significant role in both impacting human health ad-
versely (Kelly and Fussell, 2015) and influencing climate (Boucher et al., 2013). Formed in the atmosphere-mimicking gas



phase and containing six or more oxygen atoms (Bianchi et al., 2019), HOM are produced via $RO_2$ autoxidation which rapidly
increases their oxygen content through intramolecular H atom abstractions followed by $O_2$ additions (Crounse et al., 2013; Ehn
et al., 2014). Eventually, these highly oxygenated $RO_2$ will generally be terminated similarly as other $RO_2$, such as through
bimolecular reactions with $NO_x$, $RO_2$ or $HO_2$ radicals (i.e., R4, R5 and R6).

$$RO_2 + RO_2 \rightarrow ROOR + O_2 \hspace{6cm} \text{R5a}$$
$$\rightarrow ROH + RC{=}O + O_2 \hspace{5cm} \text{R5b}$$
$$RO_2 + HO_2 \rightarrow ROOH + O_2 \hspace{5.5cm} \text{R6a}$$
$$\rightarrow RC{=}O + H_2O + O_2 \hspace{4.8cm} \text{R6b}$$

The $O_3$ formation precursors, namely $NO_x$ and VOC, thus are intrinsically connected also to HOM formation through $RO_2$
chemistry. As such, if the daytime HOM distribution is dominated by organic nitrates, it suggests that the majority of $RO_2$ are
being terminated by reactions with NO (R4), thus contributing to $O_3$ formation. On the other hand, if we observe large amounts
of HOM dimers or non-nitrate monomers, formed from $RO_2$ + $RO_2$/$HO_2$ (R5 and R6), there must be a large fraction of $RO_2$
that do not contribute to the $O_3$ formation process, suggesting that increased $NO_x$ would also lead to more $O_3$. In other words,
ratios of different types of HOM can function as another indicator for determining the sensitivity of $O_3$ formation. The situation
is complicated by several factors, including challenges in identifying which HOM might have formed from $RO_2$ termination
by $RO_2$ or $HO_2$, or by knowing if a HOM is a monomeric product of a larger VOC precursor or a dimeric product from smaller
VOC. Still, if HOM could be used even as a qualitative indicator for $O_3$ formation, one particular benefit would be that they
would serve as a real-time indicator. This is because both the formation (through autoxidation) and loss (through condensation
onto aerosol particles) of HOM take place on timescales of minutes or less.

In this study, our objective is to assess the viability of using the ratio of HOM dimers or non-nitrate monomers to HOM
organic nitrates as an indicator of $O_3$ formation sensitivity. We conducted a series of experiments in an atmosphere simulation
chamber (Riva et al., 2019a), focusing on the ozonolysis of $\alpha$-pinene, the most abundantly emitted monoterpene (Pathak et al.,
2007). By varying the concentrations of $O_3$ formation precursors $NO_x$ and $\alpha$-pinene, as well as the $NO_2$ photolysis rate, we
explored the shift between $NO_x$-limited and VOC-limited regimes in the chemical system. We employed mass spectrometers
and gas monitors to measure HOM products, $\alpha$-pinene, $O_3$, and $NO_x$. We also developed a simple 0-D box model to simulate
the concentrations of $O_3$ and its precursors in the chamber under different conditions. Finally, by analyzing both experimental
and model outcomes, we evaluated the potential of the HOM ratios as indicators of $O_3$ formation sensitivity (either VOC-
limited or $NO_x$-limited) in this system.



## 2 Methods

### 2.1 Experiments

The experiments were conducted in the COALA chamber, as presented by Riva et al. (2019a). The cuboid chamber is made of fluorinated ethylene propylene (FEP) and has a volume of 2 m$^3$ with a volume to surface area ratio of 0.2. The chamber was run in "steady-state mode" (Peräkylä et al., 2020; Krechmer et al., 2020), meaning there was a continuous flow of air and reagents (O$_3$, NO$_2$, $\alpha$-pinene) through the chamber. The total flow was around 55 L min$^{-1}$, giving an average residence time of approximately 36 minutes ($\tau = \frac{2000 \text{ L}}{55 \text{ L min}^{-1}} \approx 36 \text{ min}$). Each stage, where the experimental conditions remain unchanged,

lasted at least 1.5 hours, which is approximately three times the residence time, allowing the chamber to reach a pseudo steady state, as confirmed by the time series obtained during the experiments (e.g., Fig. 4).

**Table 1.** Experimental conditions. Each experiment consisted of 3-9 "stages" that corresponded to a specific time period during which the inputs remain constant. The parameter that was varied included input O$_3$, $\alpha$-pinene, or NO$_x$ concentrations, as well as NO$_2$ photolysis rate ($J_{NO_2}$). These variations are indicated in the table by multiple values or ranges in a given cell. Experiment numbers (No.) and number of total stages per experiment are shown in the first two columns.

| Experiment *No.* | Number of stages | Input | | | |
|---|---|---|---|---|---|
| | | $J_{NO_2}$ (s$^{-1}$) | O$_3$ (ppb) | $\alpha$-pinene (ppb) | NO$_x$ range (ppb) |
| *1.* | 7 | 1.85×10$^{-3}$ | 10.5 | 30/60 | 0 - 21.5 |
| *2.* | 8 | 1.85×10$^{-3}$ | 15 | 15/45/60 | 0 - 21.5 |
| *3.* | 9 | 1.85×10$^{-3}$ | 22.5 | 10/45/60 | 0 - 35.2 |
| *4.* | 9 | 1.15×10$^{-3}$ | 15.5 | 15/45/60 | 0 - 44.5 |
| *5.* | 9 | 0.35×10$^{-3}$ | 15.5 | 15/45/60 | 0 - 44.5 |
| *6.* | 3 | 1.85×10$^{-3}$ | 10/15.5/22.5 | 60 | 44.5 |
| *7.* | 8 | 1.85×10$^{-3}$ | 10 | 30/45/60 | 0 - 21.5 |

The details of the conducted experiments are provided in Table 1. The UV LED lights (wavelength $\sim$ 400 nm, manufactured by LEDlightmake Inc., Shenzhen, China) (Zhao et al., 2023a) used for photolyzing NO$_2$ were kept on throughout each experiment, while the input concentrations of the precursors NO$_2$, O$_3$ and $\alpha$-pinene were varied across experiments and stages to

map out a wide range of different conditions. The photolysis rate was varied by using varying numbers of LED light strips (1, 3, or 5). Experiments without VOC addition (no. Z1-Z5, Table A1) were used to evaluate the photolysis rates (given in Table 1) for each number of light strips. We acknowledge that using alkene ozonolysis for this type of study is not ideal as O$_3$ also reacts with the VOC, making the determination of actual O$_3$ formation more complicated. The choice of this system was partly due to our chamber not having a good light source for producing OH radicals, thus limiting us to O$_3$ oxidation, and partly because

the HOM spectra from this system has been studied in great detail, making the interpretation of the HOM easier.





All the input reactants, as well as the HOM products, were continuously measured online using instruments described below. The identified HOM species in this study were categorized into three groups: 1) "HOM monomers" ($HOM_{Mono}$), $C_8$-$C_{10}$ compounds without any nitrogen atoms; 2) "HOM organic nitrates" ($HOM_{ON}$), $C_8$-$C_{10}$ compounds with one nitrogen atom; and 3) "HOM dimers" ($HOM_{Di}$), $C_{18}$-$C_{20}$ compounds without any nitrogen atoms.

## 2.2 Instrumentation

### 2.2.1 Mass spectrometers

A nitrate-adduct Chemical Ionization Mass Spectrometer ($NO_3$-CIMS, Tofwerk AG/Aerodyne Research, Inc.) was used for online measurements of HOM with high selectivity (Jokinen et al., 2012; Ehn et al., 2014; Riva et al., 2019b). A large sheath flow of 20 L min$^{-1}$ (to minimize wall losses) carries nitric acid ($HNO_3$) across X-rays, producing nitrate ions ($NO_3^-$). Then, in an electric field, $NO_3^-$ are guided towards a 10 L min$^{-1}$ sample flow, ionizing targeted HOM molecules by clustering with them. (Kürten et al., 2014). Finally, the charged sample molecules are directed through a critical orifice and into an Atmospheric Pressure Interface Time-of-Flight Mass Spectrometer (APi-TOF), where they are detected based on mass-to-charge ratios (m/z) (Junninen et al., 2010). The $NO_3$-CIMS was equipped with a standard TOF (HTOF), having a mass resolution of 5000 at m/z 188 Th. The concentrations of HOM were converted from their normalized signals (i.e., the ratio of HOM-containing ions to reagent ions) by multiplying with a calibration factor ($C$), which takes different efficiencies into account (Jokinen et al., 2012; Bianchi et al., 2019):

$$[HOM] = C \cdot \frac{HOM(NO_3^-)}{\sum_{i=0}^{2}(HNO_3)_i(NO_3^-)} \tag{1}$$

Calibrating with sulfuric acid (Kürten et al., 2012), we determined $C$ to be $1.56 \times 10^9$ cm$^{-3}$ ($\pm 50\%$) based on a flow-tube model (He et al., 2023). However, in this study, the accuracy of $C$ is less important since the normalized signals of HOM were sufficient for relative comparisons.

A Proton Transfer Reaction Time-of-Flight mass spectrometer (PTR-TOF 8000, Ionicon Analytik GmbH), designed for online measurements of VOC (Jordan et al., 2009), was utilized in our study specifically for the detection of $\alpha$-pinene. The PTR is an ionization method where water molecules ($H_2O$) are ionized in a hollow cathode discharge, resulting in the formation of hydronium ions ($H_3O^+$) (Hansel et al., 1995). Then, as proton donors, $H_3O^+$ are directed into a drift tube, where trace organic compounds are ionized by proton transfer process with proton affinity as the key parameter (Hansel et al., 1995; Graus et al., 2010). After a differentially pumped ion transfer unit, the charged molecules enter a TOF, where collisions are negligible under a low pressure of $\sim 10^{-6}$ mbar with a high vacuum (Graus et al., 2010). The inlet flow was 1 L min$^{-1}$ with 0.1 L min$^{-1}$ being sampled into the ion drift tube. Further details regarding the calibration and settings of the PTR can be found in Zhao et al. (2023b), and the calibration factor for $\alpha$-pinene (detected as $C_{10}H_{17}^+$ at m/z 137 Th) was $\sim 104$ ppb$^{-1}$ after normalization by primary ion isotope $H_3^{18}O^+$ (at 21 Th). The analysis of raw data from the $NO_3$-CIMS and PTR-TOF was conducted using the MATLAB-based set of programs called tofTools (version 611) (Junninen et al., 2010).





### 2.2.2 Gas monitors

The concentrations of $NO_x$ and $O_3$ were measured by gas monitors. A photometric $O_3$ analyzer - model 400 (Teledyne API) was used to detect $O_3$ in the chamber. The amount of $O_3$ determines how much of a 254 nm UV light signal is absorbed in the sample cell. The absorption difference between the intact sample air and the $O_3$-removed air, achieved by a switching valve periodically, enables the determination of the stable $O_3$ concentrations.

As for $NO_x$, a NO-$NO_2$ analyzer - model T200UP (Teledyne API) was utilized. With a high-efficiency photolytic converter, $NO_2$ is transformed to NO with minimal interference from other gases. Using the chemiluminescence detection principle, NO is measured by reacting with $O_3$, yielding light in direct proportion to the amount of NO (Archer et al., 1995). In this way, both sampled NO and total $NO_x$ can be measured, without and with using the photolytic converter, respectively. This enables the determination of the $NO_2$ concentration in the sample by subtraction.

### 2.3 Box model

As $O_3$ was both injected and produced in the chamber, and sinks included reactions with NO and VOC as well as flush out, we constructed a simple 0-D box model (12 reactions, 9 species, Table A2) to mimic the main reactions and to generate $O_3$ isopleth diagrams. These isopleths were then used to determine sensitivity regimes for $O_3$ formation. Reaction rates were adapted from the NIST Chemical Kinetics Database (https://kinetics.nist.gov/kinetics/index.jsp). The model did not include closed-shell HOM products and all peroxy radicals were treated as a single term (i.e., "$RO_2$"). The box model was first employed to determine $NO_2$ photolysis rates under different numbers of UV lights in the zero-VOC experiments (Table A1), where only $O_3$ and $NO_2$ were used as input species. The high agreement between model and observation (detailed results are shown in section 3.1 and 3.3) showed that the model was adequate for simulating the targeted reactions in our chamber.





# 3 Results and discussions

## 3.1 NO₂ photolysis rate determination

Figure 1 illustrates an example of a zero-VOC experiment, while Table A1 provides a comprehensive list of all zero-VOC experiments conducted. Using the box model, NO₂ photolysis rates ($J_{NO_2}$) were fitted to accurately simulate the gas concen-

trations, namely $O_3$ and $NO_x$, at each stage of the experiment. With a NO₂ photolysis rate of $1.85 \times 10^{-3}$ s$^{-1}$ with 5 UV lights, the modeled gas concentrations agreed extremely well with measured values (Fig. 1). Similar agreement was observed for different inputs of $O_3$ and $NO_2$ (Fig. A2 and A3), indicating the robustness of both model and fitted $J_{NO_2}$. Furthermore, employing the same procedure, the photolysis rates with 3 and 1 UV lights were determined to be $1.15 \times 10^{-3}$ s$^{-1}$ and $0.35 \times 10^{-3}$ s$^{-1}$, respectively (Fig. A4 and A5).

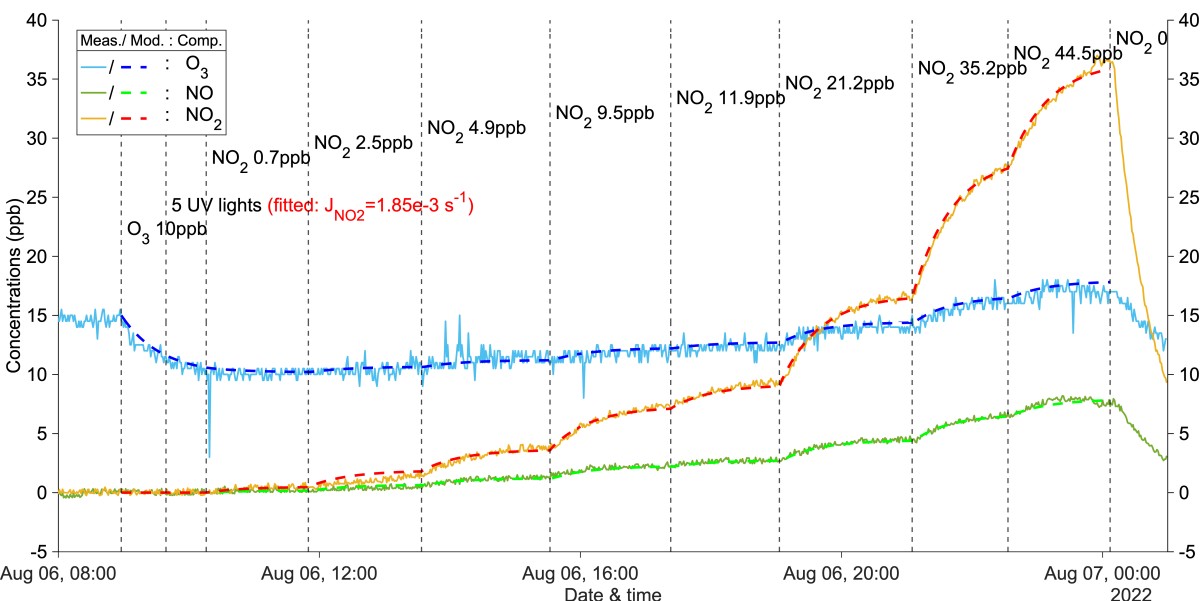

**Figure 1.** A zero-VOC experiment (Z1) for determining the photolysis rate of NO₂. Measured (abbreviated as Meas.) and modeled (abbreviated as Mod.) concentrations of different compounds (abbreviated as Comp.) are shown in solid and dashed lines respectively. Dashed vertical lines indicate specific time points of operations, with corresponding labels for each operation. Note that the operation labels show input information (input NO₂ = measured NO₂ + NO).

## 195    3.2 HOM determination

A steady-state spectrum (experiment no. 2) obtained from the NO₃-CIMS (Fig. 2) illustrates the identified closed-shell HOM products, including HOM$_{Mono}$ (light green), HOM$_{ON}$ (blue), and HOM$_{Di}$ (dark green). Table 2 provides the formulas of all the selected HOM species. Figure A6 displays the steady-state spectra of all stages, with the corresponding input information



described in Fig. 4a. As expected, with the injection of more $NO_2$, the signals of $HOM_{ON}$ increased, while those of $HOM_{Mono}$

and $HOM_{Di}$ decreased considerably (stage (2) - (5) in Fig. A6). This observation is consistent with the dominance of the $RO_2$ + NO reaction over the $RO_2$ cross-reactions at a few ppb of $NO_x$ (Yan et al., 2016, 2020). After the addition of more $\alpha$-pinene, all signals showed a noticeable rise (stage (6) - (7) in Fig. A6) due to the increased supply of $RO_2$ species.

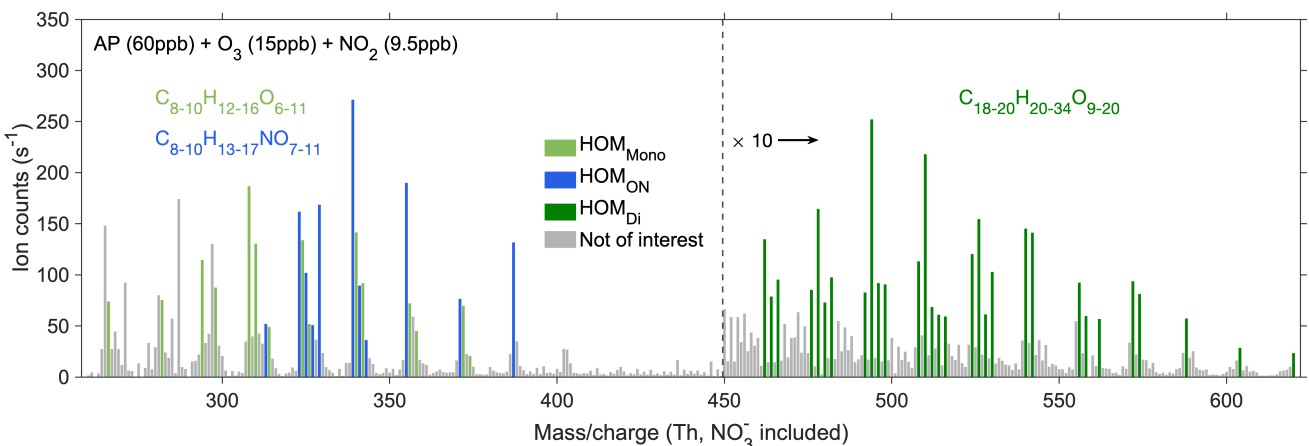

**Figure 2.** The steady-state spectrum (15 min average) at the stage (7) of experiment no. 2 from the $NO_3$-CIMS. The spectrum was corrected by subtracting the corresponding background signals. Light green bars show $HOM_{Mono}$, dark green ones show $HOM_{Di}$, blue ones show $HOM_{ON}$ while grey ones show peaks not of interest. The peaks not of interest either exhibited relatively low signals, represented radicals, contained too small a number of carbon/oxygen atoms, or had uncertain mass-to-charge ratios. The peaks larger than 450 Th are multiplied by 10.

**Table 2.** Identified HOM closed-shell species based on the experiment no. 2 The reagent ion $NO_3^-$ is excluded.

| Monomers | Dimers | |
|---|---|---|
| $C_8H_{12}O_x$ (x=6-9) | $C_{18}H_{22}O_x$ (x=17-20) | $C_{19}H_{32}O_x$ (x=10-13) |
| $C_{10}H_{14}O_x$ (x=7-11) | $C_{18}H_{24}O_x$ (x=12,14,16,17) | $C_{20}H_{20}O_{15}$ |
| $C_{10}H_{16}O_x$ (x=6-11) | $C_{18}H_{26}O_x$ (x=11-14) | $C_{20}H_{30}O_x$ (x=10-18) |
| **Organic nitrates** | $C_{18}H_{26}O_x$ (x=11-14) | $C_{20}H_{32}O_x$ (x=9,11-15) |
| $C_8H_{13}NO_x$ (x=8,9) | $C_{18}H_{30}O_{13}$ | $C_{20}H_{34}O_x$ (x=11,12) |
| $C_9H_{15}NO_x$ (x=8,9) | $C_{19}H_{26}O_x$ (x=10-19) | |
| $C_{10}H_{15}NO_x$ (x=7-11) | $C_{19}H_{28}O_x$ (x=9-16) | |
| $C_{10}H_{17}NO_x$ (x=7,8) | $C_{19}H_{30}O_x$ (x=9-13) | |

Our experiments showed that HOM organic nitrates with fewer than 9 oxygen atoms ($HOM_{ON,O\leq8}$) exhibited the slowest decay at the end of the experiment (Fig. 3). This can be explained by the evaporation of these semi-volatile $HOM_{ON,O\leq8}$





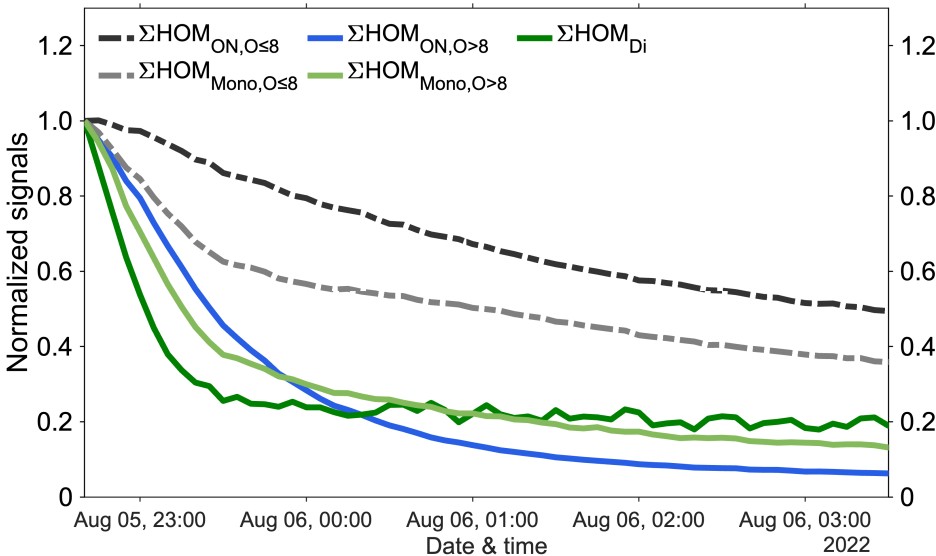

**Figure 3.** The normalized signal decay of HOM$_{ON}$, HOM$_{Mono}$, and HOM$_{Di}$ after experiment no. 2 ended. All signals were normalized first to primary ions and then to the signals at the moment when experiment no. 2 ended. $\sum$HOM$_{ON,O>8}$ (sum of HOM$_{ON}$ species with more than 8 oxygen atoms) are in solid blue lines, $\sum$HOM$_{ON,O\leq8}$ are in dashed black lines, $\sum$HOM$_{Mono,O>8}$ are in solid light green lines, $\sum$HOM$_{Mono,O\leq8}$ are in dashed grey lines while $\sum$HOM$_{Di}$ are in solid dark green lines.

compounds from chamber walls even after the gas phase production had stopped. Despite showing a faster decay compared to HOM$_{ON,O\leq8}$, non-nitrate HOM monomers with fewer than 9 oxygen atoms (HOM$_{Mono,O\leq8}$) also showed overall slow decays (Fig. 3). As a result, prior to subsequent experiments, the concentration levels of HOM$_{ON,O\leq8}$ and HOM$_{Mono,O\leq8}$ remained high, obscuring actual concentration changes following the addition of NO$_x$. In contrast, HOM$_{ON,O>8}$, HOM$_{Mono,O>8}$ and HOM$_{Di}$ experienced fast decays, reaching a low and stable level ($\leq0.2$) within a few hours after the experiment (Fig. 3). Despite the higher normalized background level of HOM$_{Di}$ due to their lower concentrations, this stable level allowed for accurate background subtraction in subsequent experiments, enabling measurements of true changes in concentration. Therefore, to ensure the accuracy of our results, we excluded the less oxygenated and more volatile HOM species, specifically HOM$_{ON,O\leq8}$ and HOM$_{Mono,O\leq8}$. Consequently, the indicating ratio used in this study is defined as the ratio between the sum of HOM$_{Di}$ or HOM$_{Mono,O>8}$ and the sum of HOM$_{ON,O>8}$, represented as $\frac{\sum\text{HOM}_{Di}}{\sum\text{HOM}_{ON,O>8}}$ (Indicating Ratio 1, abbreviated as IR1) or $\frac{\sum\text{HOM}_{Mono,O>8}}{\sum\text{HOM}_{ON,O>8}}$ (IR2).

### 3.3 Indicating ratios

In this section, we detail the conducted experiments. Experiment no. 2, is depicted in Fig. 4, while the other experiments are shown in Fig. A7-A12. Variation of NO$_2$ and $\alpha$-pinene input concentrations lead to changes in both indicating ratios (IR1 and



IR2) that correspond with changes in $O_3$ concentrations (Fig. 4), suggesting a possible sensitivity of $O_3$ formation. From stage
(1) to (5), injection of more $NO_2$ led to increased formation of $HOM_{ON,O>8}$, while production of $HOM_{Di}$ and $HOM_{Mono,O>8}$
was suppressed. As a result, the indicating ratios decreased. Additionally, the concentration of $O_3$ increased during this period,
but the rate of increase decreased with higher $NO_2$ inputs. This trend suggests a gradual shift from a $NO_x$-limited regime to
a more VOC-limited regime in the system. For example, during stage (5) with 9.5 ppb $NO_2$, the $O_3$ concentration remained
relatively constant. This observation indicates the system may have shifted to the VOC-limited regime. Next, when additional
$\alpha$-pinene was introduced ($\sim$ 45 ppb during stage (6)), a significant increase in $O_3$ concentration was again observed, consistent
with the system being in the VOC-limited regime during stage (5). However, after injection of $\sim$ 60 ppb $\alpha$-pinene during stage
(7), the $O_3$ concentration reached a plateau, indicating that the system had shifted back to the $NO_x$-limited regime. Moreover,
during these two stages, the indicating ratios experienced a substantial increase. In the last stage (8), when additional $NO_2$ was
injected to reach 21.5 ppb, while input $\alpha$-pinene concentration remained unchanged, the $O_3$ concentration increased again, and
the indicating ratios decreased noticeably, confirming that the previous stage was $NO_x$-limited. However, it should be noted
that the system in this stage might not yet be VOC-limited.

Other experiments with 5 UV lights (Fig. A7, A8, and A11) also exhibited similar time series patterns as described above.
One noticeable difference is that higher initial $O_3$ concentrations resulted in less pronounced increases in $O_3$ during the first
5 stages with varying $NO_2$ levels. This can be attributed to the reaction $O_3$ + NO (R2) becoming faster, competing with the
formation of $O_3$ from the $RO_2$ + NO reaction (R4a) followed by $NO_2$ photolysis (R1). As a result, there was a reduced $O_3$
formation in the presence of higher initial $O_3$ concentrations due to the scavenging of NO by existing $O_3$.

Compared to experiment no. 2 (Fig. 4), experiments with fewer lights (but similar initial concentrations; Fig. A9 and A10)
provide insight into the effect of UV light intensities on $O_3$ formation sensitivity and the consistency of the indicating ratios
under different light intensities. More lights led to more pronounced increase in $O_3$ concentrations at the same stages because
additional $O_3$ was produced from $NO_2$ photolysis. On the other hand, fewer lights resulted in lower NO levels in the system
since $NO_2$ input was the sole source of $NO_x$. This led to reduced $NO_2$ formation from $RO_2$ + NO reaction (R4a), and subse-
quently less $O_3$ formation. This aspect is crucial in determining the $O_3$ sensitivity. In this sense, the presence of fewer lights
implies that higher levels of $NO_2$ input are required to ensure sufficient NO levels for reaching the VOC-limited regime. This
can be confirmed by the stage where $\sim$ 45 ppb $\alpha$-pinene was injected (Fig. 4, A9 and A10). During this stage, the $O_3$ con-
centration showed a significant increase with either 3 or 5 UV lights, indicating that the system had reached the VOC-limited
regime in the previous stage. However, when only 1 UV light was used, the $O_3$ concentration remained relatively constant,
suggesting that the system with 1 light did not reach the VOC-limited regime. It is worth mentioning that the indicating ratios
exhibited more significant changes when using more lights. This can be attributed to the fact that more lights result in increased
production of $O_3$ and NO, leading to more drastic changes in HOM distributions and thereby influencing the indicating ratios
to a greater extent.

Using the box model described in section 2.3, the concentrations of $O_3$ and its precursors were captured well both qual-
itatively and quantitatively (e.g., Fig. 4a). In general, simulated concentrations of $O_3$ during the steady states differed from
measured values by at most 10%, while differences for $NO_x$ were even smaller. The largest discrepancy in concentrations



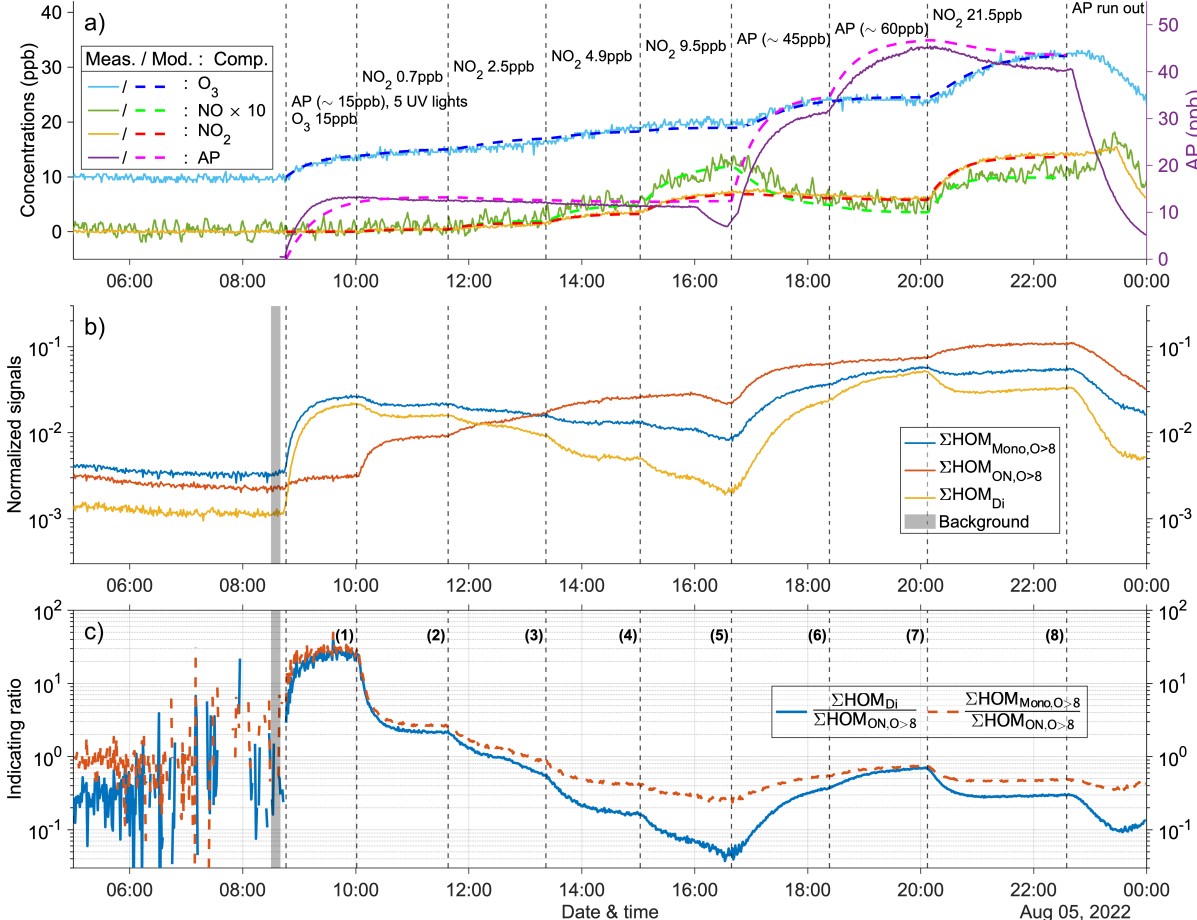

**Figure 4.** Time series of experiment no. 2 with 15 ppb $\alpha$-pinene and 15 ppb $O_3$ as initial inputs. 5 UV lights were on during all stages. Three subplots show the time series of different compounds: a) measured (abbreviated as Meas., in solid lines) and modeled (abbreviated as Mod., in dashed lines) concentrations of $O_3$, $NO_x$ (both shown by left y-axis; the NO concentration is multiplied by 10), and AP (i.e., $\alpha$-pinene, shown by right y-axis); b) normalized signals of $\sum HOM_{Mono,O>8}$ (sum of non-nitrate HOM monomers with more than 8 oxygen atoms), $\sum HOM_{ON,O>8}$ (sum of HOM organic nitrates with more than 8 oxygen atoms), and $\sum HOM_{Di}$ (sum of HOM dimers); c) IR1 ($\frac{\sum HOM_{Di}}{\sum HOM_{ON,O>8}}$) and IR2 ($\frac{\sum HOM_{Mono,O>8}}{\sum HOM_{ON,O>8}}$). The grey shaded area represents the time period selected for background subtraction before calculating the ratio. Dashed vertical lines indicate specific time points of operations, with the corresponding labels for each operation in the subplot a. The bolded number in parentheses in subplot c corresponds to the number of stages. The steady-state mass spectra obtained by the $NO_3$-CIMS for each stage is shown in Fig. A6.

($\sim 15\%$) was observed for $\alpha$-pinene, which can be attributed to the simplifications made in the model. Specifically, the OH
concentration will be underestimated if the model did not accurately capture the yields of $HO_2$, which can be converted into OH via reactions with NO.





Overall, both indicating ratios are promising as indicators of $O_3$ formation sensitivity. However, in all time series (Fig. 4c, A7c-A12c), IR1 ($\frac{\sum HOM_{Di}}{\sum HOM_{ON,O>8}}$) exhibited more pronounced changes compared to IR2 ($\frac{\sum HOM_{Mono,O>8}}{\sum HOM_{ON,O>8}}$) as we shifted the $O_3$ formation regimes. This highlights that IR1 holds better potential for quantitatively indicating $O_3$ formation sensitivity in the absence of other perturbing factors. The intensity of light (i.e., $NO_2$ photolysis rate) played a crucial role in determining $O_3$ sensitivity regimes by controlling the amount of NO. Our box model successfully reproduced the measured values of $O_3$ and its formation precursors, making it reasonable to extend the model to generate $O_3$ isopleths under chamber conditions beyond those covered in our experiments. With these we can better elucidate how well the HOM ratios can function as $O_3$ sensitivity indicators in this system.

## 3.4 Viability of the indicating ratios

In order to validate the indicating ratio, we generated $O_3$ isopleths by modeling various combinations of input $NO_2$ and $\alpha$-pinene concentrations. The indicating ratios obtained from all steady state stages were scattered on the same coordinate system (IR1: Fig. 5-6 and IR2: Fig. A13-A14). The purpose of varying the injection rates of either $NO_x$ or $\alpha$-pinene within each experiment (e.g., colored curves in Fig. 5) was to shift the system between the VOC-limited and $NO_x$-limited regimes. Unlike previous studies that used the concentration of VOC as the x-axis (Kinosian, 1982; Chameides et al., 1992), in this study, the x-axis represented the product of the measured $\alpha$-pinene and $O_3$ concentrations. This is because the product reflects the potential for $RO_2$ formation, which reacts directly with NO to contribute to $O_3$ accumulation (R1-R4a). In typical EKMA plots the oxidant is primarily thought to be OH, the concentration of which is independent of the VOC concentrations, thus the x-axis would be largely equivalent regardless of whether plotting VOC or VOC times oxidant. In our case, the VOC concentration will directly influence the oxidant (i.e., $O_3$) concentration through chemical reactions, and therefore we chose to use the current x-axis. The ridge line of the EKMA curves was plotted for each experiment, represented by dotted lines in e.g., Fig. 5b-d. The reason for separating the b-d plots from each other is due to the differences in constant $O_3$ inputs or $NO_2$ photolysis rates in each experiment, resulting in distinct $O_3$ isopleths. For the context of this work, we do not separate a transition regime where $O_3$ formation is sensitive to both $NO_x$ and VOC, but simply define the VOC- and $NO_x$-limited regimes based on the ridge line. Farther above the ridge line, the system is more VOC-limited, while farther below the line, the system is more $NO_x$-limited. It's notable that in the VOC-limited regime, instead of a reduction, the $O_3$ concentration even increased slowly with the addition of $NO_x$, primarily due to the photolysis of the input $NO_2$.

Comparing experimental and model results, the steady state stages of all experiments with 5 UV lights exhibited a consistent pattern for the indicating ratios, allowing qualitative determination of $O_3$ formation sensitivity (Fig. 5 and A13). Generally, the farther the steady state point is from the ridge line, the darker the coloring (either blue or green). Specifically, when the color is darker blue, it indicates a smaller value of the indicating ratios, suggesting a higher likelihood of the system being in the VOC-limited regime. Conversely, when the color is darker green, it signifies a higher value of the ratios, indicating a higher likelihood of the system being in the $NO_x$-limited regime. It is worth noting that experiment no. 7 was essentially a duplicate of experiment no. 1, but with additional stages. When comparing the indicating ratios at stages with the same inputs, as shown in Fig. 5d and A13d, it becomes apparent that the values are highly consistent and closely aligned. Moreover, the background





**Figure 5.** Steady state IR1 ($\frac{\sum \text{HOM}_{\text{Di}}}{\sum \text{HOM}_{\text{ON,O>8}}}$) of experiments from four days with 5 UV lights. X-axis is the multiplication of measured (steady state) $\alpha$-pinene and $O_3$ concentrations, while y-axis is the measured (steady state) $NO_x$. The scatter points (exp. no. 1 (experiment number 1): diamond; exp. no. 2: star; exp. no. 3: round; exp. no. 7: square) are colored by values of IR1 (abbreviated as R. in the figure), and are connected by curves (exp. no. 1: blue; exp. no. 2: orange; exp. no. 3: green; exp. no. 7: purple) showing the sequence (Seq.) of experimental stages. Panel a combines stages of all four days, and the rest three subplots respectively show the stages of experiments with different initial (Ini.) inputs (exp. no. 1 and 7 are in the same panel d due to same initial inputs). EKMA curves (isopleth of $O_3$ concentrations in ppb), simulated by the box model, are black solid lines, while dotted lines are corresponding ridge lines.

signals of HOM$_{\text{ON,O>8}}$, HOM$_{\text{Mono,O>8}}$ and HOM$_{\text{Di}}$ increased substantially with experiments going on, from $5\times10^{-4}$ - $2\times10^{-3}$ (Fig. A7b) to $4\times10^{-3}$ - $2\times10^{-2}$ (Fig. A11b). However, the accumulating background did not have a significant impact on the







**Figure 6.** Steady state IR1 ($\frac{\sum \mathrm{HOM_{Di}}}{\sum \mathrm{HOM_{ON,O>8}}}$) of experiments from three days with 5, 3, and 1 UV lights, respectively. X-axis is the multiplication of measured (steady state) $\alpha$-pinene and $O_3$ concentrations, while y-axis is the measured (steady state) $NO_x$. The scatter points (exp. no. 2 (experiment number 2): star; exp. no. 4: diamond; exp. no. 5: round) are colored by values of IR1 (abbreviated as R. in the figure), and are connected by curves (exp. no. 2: orange; exp. no. 4: blue; exp. no. 5: green) showing the sequence (Seq.) of experimental stages. Panel a combines stages of all three days with the same initial (Ini.) inputs, and the rest three subplots respectively show the stages of experiments with different amount of UV lights. EKMA curves by the box model are in black lines and the dotted lines are corresponding ridge lines.

indicating ratios, after the background subtraction (Fig. 5d and A13d). These highlight the remarkable reproducibility of the indicating ratios in our chamber experiments.

To investigate the impact of light intensities, a similar comparison was conducted for experiments with the same initial inputs using 5, 3, and 1 UV lights (Fig. 6 and A14). The observed changes in the pattern of the indicating ratios are in line with those





observed in the 5 UV lights experiments described above. The most significant observation is that at lower UV light intensity, the ridge line shifted towards higher $NO_x$ levels at the same potential for $RO_2$ formation (i.e., the same value of x-axis) (Fig. 6b-d and A14b-d). This finding aligns with the time-series comparison presented in section 3.3, which indicates that at weaker UV

intensity, higher input $NO_2$ is required to generate NO levels sufficient to shift the system towards the VOC-limited regime. When comparing stages with the same conditions except different UV lights, we observed that generally the lower intensity of UV lights corresponded to higher values of the indicating ratios (Fig. 6, A14), showing a higher likelihood of the system being in the $NO_x$-limited regime. This finding is in agreement with the shift of the ridge line. The correlation between UV intensities, the indicating ratios, and the position of the ridge line reinforces the relationship between the indicating ratios and

$O_3$ formation sensitivity. It suggests that in addition to the relative changes, the absolute values of the indicating ratios are also informative.

The conclusion can be drawn that the indicating ratios can predict the $O_3$ formation sensitivity, both qualitatively and even quantitatively. More specifically, regardless of the light intensity, IR1/IR2 consistently indicate the VOC-limited regime when below 0.2/0.4, and the $NO_x$-limited regime when above 0.5/0.7.

**3.5 Implication and further improvements**

Our chamber study (section 3.4) confirmed the significant role of the indicating ratios in determining $O_3$ formation sensitivity, both qualitatively and quantitatively. However, in the real atmosphere, the conditions will vary significantly more than in our simple system. Most importantly, the amount of different precursor VOC will be vastly greater, and in reactions with OH, the distribution of different types of $RO_2$ radicals will also be far more complex. As an example, in a rural setting there can be

a wide variety of small VOC (C1-C4), isoprene (C5), aromatics (C6-C9) and monoterpenes (C10) that all produce significant amounts of $RO_2$. Consequently, from these molecules, dimers can form with any carbon number between 2 and 20, while monomers can have carbon numbers between 1 and 10. In addition, both VOC and oxidant concentrations as well as radiation and meteorological patterns will vary over time. This means that it will be difficult to find universal compounds to include in the $HOM_{Di}$, $HOM_{Mono}$ and $HOM_{ON}$ groups used to calculate the indicating ratios for a given site and a given time. For

example, it is likely that there will be few environments where C10-$RO_2$ from monoterpenes would be efficiently reacting with each other during daytime, meaning that the C20 dimers used here are unlikely to be usable. It remains to be seen whether the $HOM_{Mono}$-to-$HOM_{ON}$ ratio can be used for C10 compounds in areas with high monoterpene emissions. Nevertheless, conceptually the link between HOM formation pathways and $O_3$ formation should hold, and it may be possible to determine suitable compound groups for various sites. Our study focused exclusively on $\alpha$-pinene, but the intrinsic connection between

the indicating ratios we proposed and $O_3$ formation (Fig. A1) is not limited to this specific VOC. Further laboratory and ambient studies are necessary to investigate additional VOC of interest and expand our understanding of the indicating ratios' applicability and generalizability in predicting $O_3$ formation sensitivity under various atmospheric conditions. In addition, at the extremes, ranging from clear formation of dimeric species to complete lack of dimeric species with abundant organic HOM nitrates, this can be considered a strong qualitative indicator being in a $NO_x$- or VOC-limited regime, respectively.




## 4 Conclusions

Both $O_3$ and HOM are of significant interest, given their impacts from small-scale personal health to large-scale global climate. Due to the intrinsic connection between the formation mechanisms of $O_3$ and HOM, we suggest new indicators, denoted as $\frac{\sum \text{HOM}_{\text{Di}}}{\sum \text{HOM}_{\text{ON,O>8}}}$ and $\frac{\sum \text{HOM}_{\text{Mono,O>8}}}{\sum \text{HOM}_{\text{ON,O>8}}}$, for determining $O_3$ formation regimes based on the distribution of HOM. One main improvement of using HOM-based indicating ratios compared to those suggested earlier would be the short lifetimes of HOM, which means that these new indicators would be real-time indicators of the formation regime. To assess the viability of the indicating ratios, a series of chamber experiments were carried out using a $NO_3$-CIMS, a PTR-TOF, and $O_3$/$NO_x$ monitors. As expected, an increase in $NO_x$ inputs led to an increase in $HOM_{ON}$ and a decrease in $HOM_{Di}$, $HOM_{Mono}$ and the indicating ratios. Conversely, an increase in $\alpha$-pinene resulted in a rise in the indicating ratios. Furthermore, when adding enough of one of the $O_3$ formation precursors (either $NO_x$ or $\alpha$-pinene), the rate of increase in $O_3$ concentration slowed down or even stopped. This indicates that the system was shifting to or had already reached the other limited regime.

With a box model, which closely reproduced the measured concentrations of $O_3$ and its precursors, $O_3$ isopleths were obtained for different concentrations of $NO_x$ and $\alpha$-pinene. After drawing ridge lines of the isopleths, it was observed that the indicating ratios provide a qualitative prediction of the $O_3$ formation regimes: lower values of the ratios indicate a greater likelihood of the system being located in the VOC-limited regime, and vice versa. With less intense UV light ($\lambda \approx 400$ nm), a higher amount of $NO_2$ was required to shift the system towards the VOC-limited regime. This can be attributed to a decrease in the formation of NO from $NO_2$ photolysis. Nevertheless, the absolute values of the indicating ratios exhibited a consistent behavior across different intensities of UV light, suggesting that these absolute values are highly valuable for analyzing $O_3$ formation sensitivity.

The main objective of this study was to evaluate the concept of using HOM distributions in indicating $O_3$ formation sensitivity. Based on our outcomes, we can conclude that the ratio of HOM dimers or non-nitrate monomers to HOM organic nitrates (i.e., $\frac{\sum \text{HOM}_{\text{Di}}}{\sum \text{HOM}_{\text{ON,O>8}}}$ or $\frac{\sum \text{HOM}_{\text{Mono,O>8}}}{\sum \text{HOM}_{\text{ON,O>8}}}$) has the capability to indicate $O_3$ formation sensitivity, even quantitatively, in this simple system of monoterpene ozonolysis. An indicating ratio of this kind would aid in better control of $O_3$ pollution and have the potential to be incorporated as a useful parameter in global models for analyzing $O_3$ formation sensitivity under diverse environmental conditions.

Nevertheless, future studies will need to assess whether this approach is feasible to be applied in real-world conditions where the chemistry is far more complex. The variability of VOC precursors alone will greatly perturb the ideal situation observed in our chamber. Still, we posit that environments with high monoterpene emissions will also produce abundant C10-$RO_2$ concentrations, and the comparison of the highly oxygenated monomeric termination products (i.e., nitrates vs non-nitrates) can provide an indication of the relative $RO_2$ termination pathways. Further studies, both ambient observation as well as chamber experiments involving multiple VOC and oxidants, will be necessary to determine the potential of HOM-based indicators for $O_3$ formation.





*Code availability.* Code is available upon request from the corresponding authors.

*Data availability.* Data is available upon request from the corresponding authors.





## Appendix A: Additional figures and tables

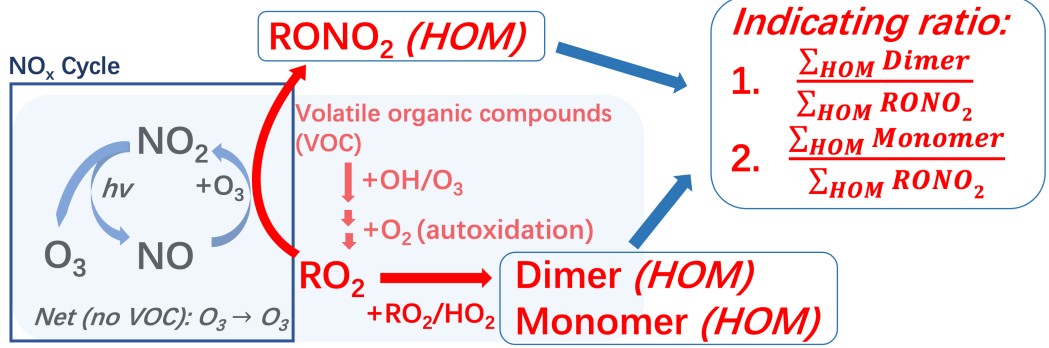

**Figure A1.** Sketch of the connection between HOM and $O_3$ formation. Based on the formation connection, two indicating ratios between HOM species are defined. "RONO$_2$ *(HOM)*": nitrate-containing HOM monomers, "Dimer *(HOM)*": non-nitrate HOM dimers, "Monomer *(HOM)*": non-nitrate HOM monomers.

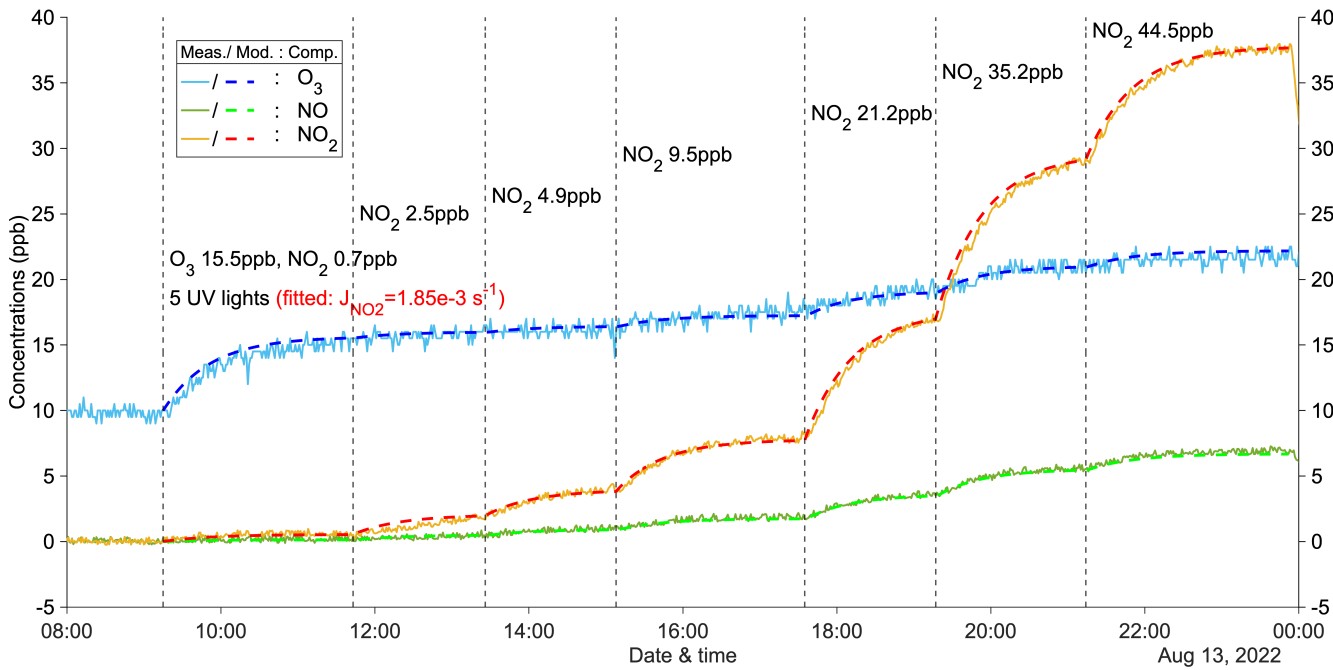

**Figure A2.** A zero-VOC experiment (Z2) for determining the photolysis rate of NO$_2$. Measured (abbreviated as Meas.) and modeled (abbreviated as Mod.) concentrations of different compounds (abbreviated as Comp.) are shown in solid and dashed lines respectively. Dashed vertical lines indicate specific time points of operations, with corresponding labels for each operation.





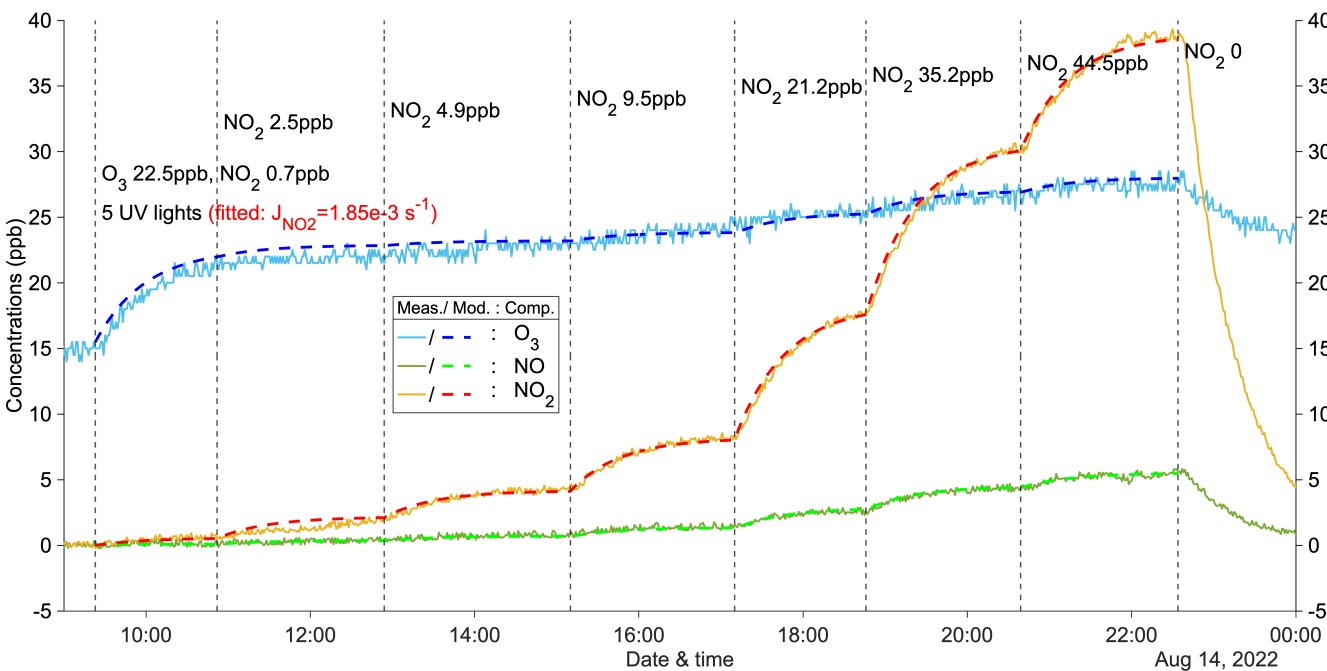

**Figure A3.** A zero-VOC experiment (Z3) for determining the photolysis rate of NO₂. Measured (abbreviated as Meas.) and modeled (abbreviated as Mod.) concentrations of different compounds (abbreviated as Comp.) are shown in solid and dashed lines respectively. Dashed vertical lines indicate specific time points of operations, with corresponding labels for each operation.



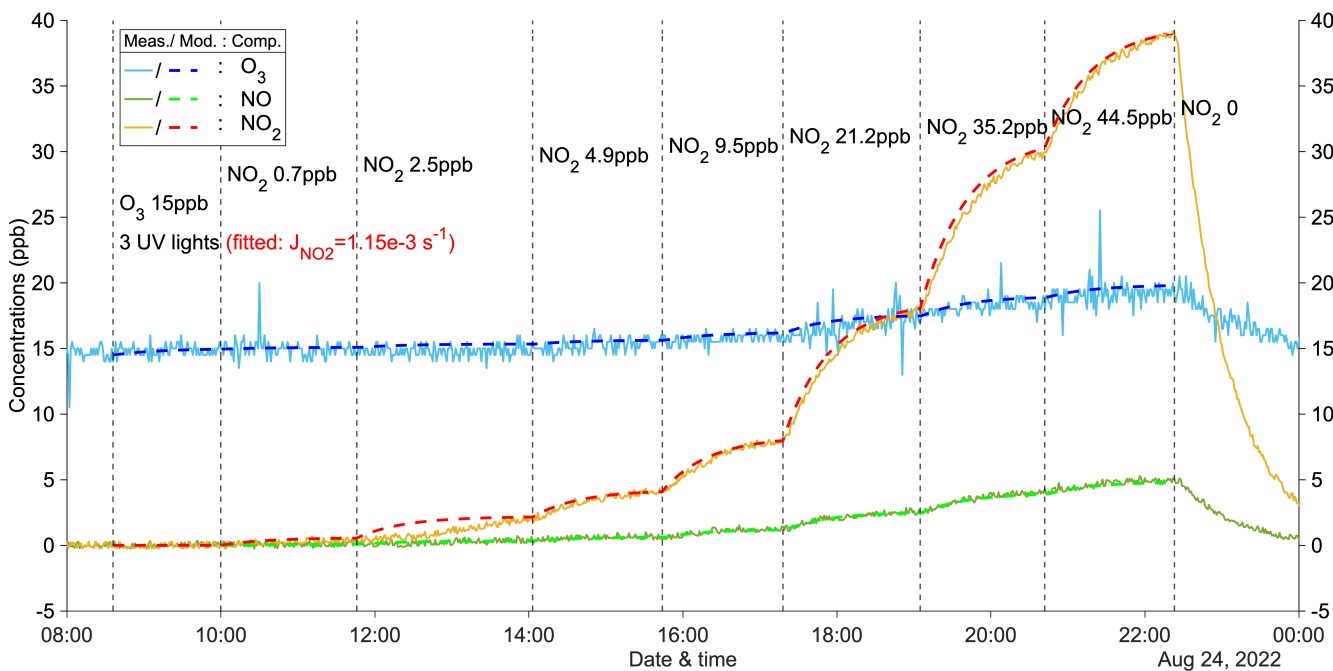

**Figure A4.** A zero-VOC experiment (Z4) for determining the photolysis rate of $NO_2$. Measured (abbreviated as Meas.) and modeled (abbreviated as Mod.) concentrations of different compounds (abbreviated as Comp.) are shown in solid and dashed lines respectively. Dashed vertical lines indicate specific time points of operations, with corresponding labels for each operation.

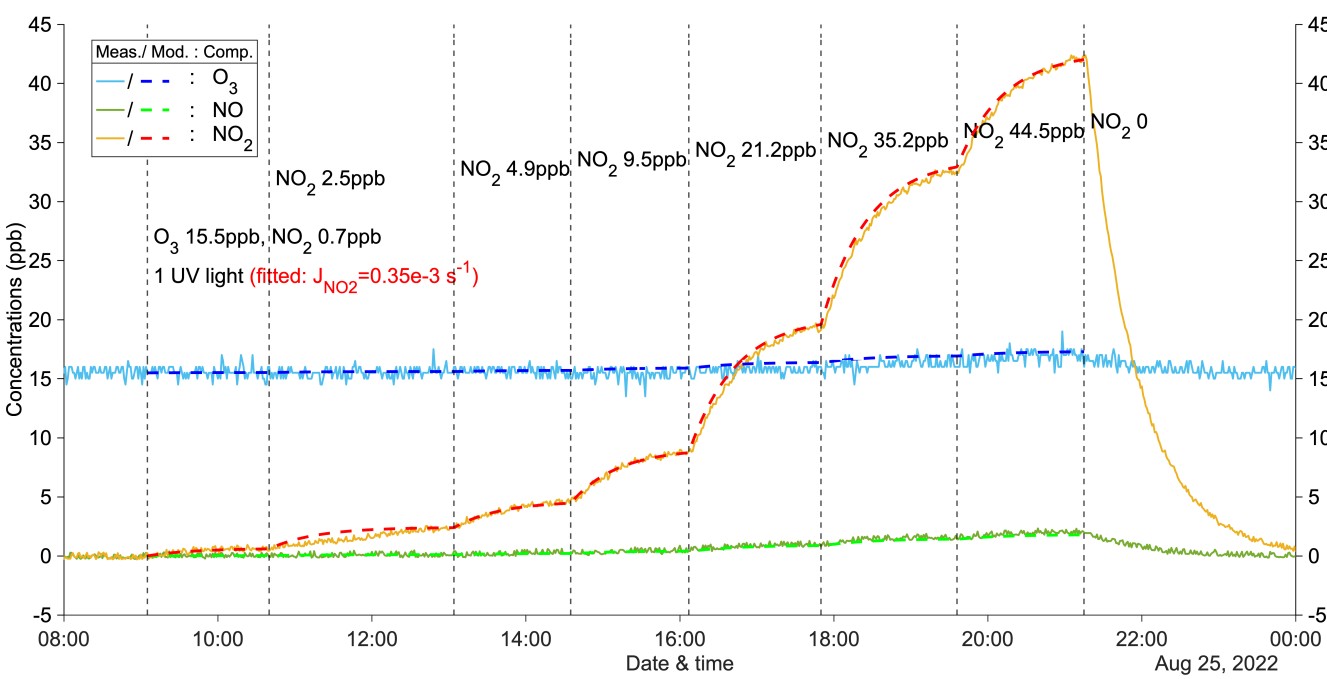

**Figure A5.** A zero-VOC experiment (Z5) for determining the photolysis rate of NO₂. Measured (abbreviated as Meas.) and modeled (abbreviated as Mod.) concentrations of different compounds (abbreviated as Comp.) are shown in solid and dashed lines respectively. Dashed vertical lines indicate specific time points of operations, with corresponding labels for each operation.



**Figure A6.** Steady-state spectra (15 min average) at experiment no. 2 from the NO$_3$-CIMS. All spectra were corrected by subtracting the corresponding background signals. The number at each row show the order of the stage, consistent with the time series in Fig. 4. Light green bars show HOM$_{Mono}$, dark green ones show HOM$_{Di}$, blue ones show HOM$_{ON}$ while grey ones show peaks not of interest. The peaks larger than 450 Th are multiplied by 10.

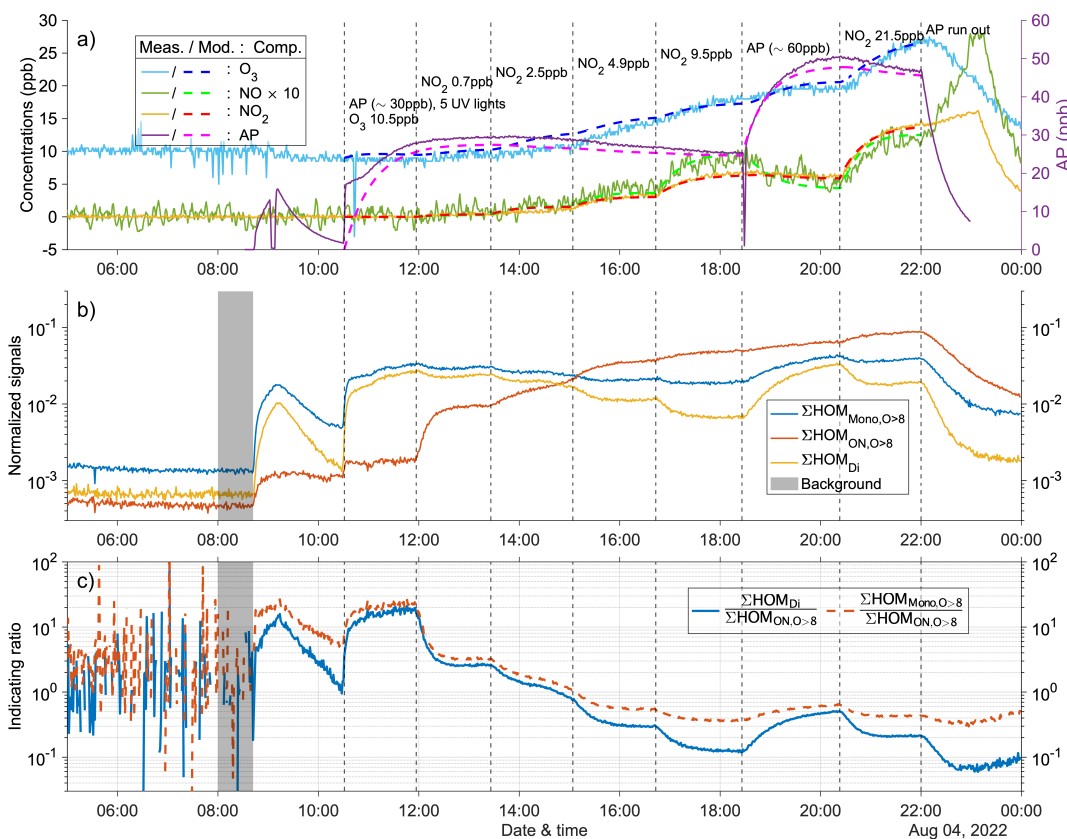

**Figure A7.** Time series of experiment no. 1 with 30 ppb $\alpha$-pinene and 10.5 ppb $O_3$ as initial inputs. 5 UV lights were on during all stages. Three subplots show the time series of different compounds: a) measured (abbreviated as Meas., in solid lines) and modeled (abbreviated as Mod., in dashed lines) concentrations of $O_3$, $NO_x$ (both shown by left y-axis; the NO concentration is multiplied by 10), and AP (i.e., $\alpha$-pinene, shown by right y-axis); b) normalized signals of $\sum HOM_{Mono,O>8}$ (sum of non-nitrate HOM monomers with more than 8 oxygen atoms), $\sum HOM_{ON,O>8}$ (sum of HOM organic nitrates with more than 8 oxygen atoms), and $\sum HOM_{Di}$ (sum of HOM dimers); c) IR1 ($\frac{\sum HOM_{Di}}{\sum HOM_{ON,O>8}}$) and IR2 ($\frac{\sum HOM_{Mono,O>8}}{\sum HOM_{ON,O>8}}$). The grey shaded area represents the time period selected for background subtraction before calculating the ratio. Dashed vertical lines indicate specific time points of operations, with the corresponding labels for each operation in the subplot a.





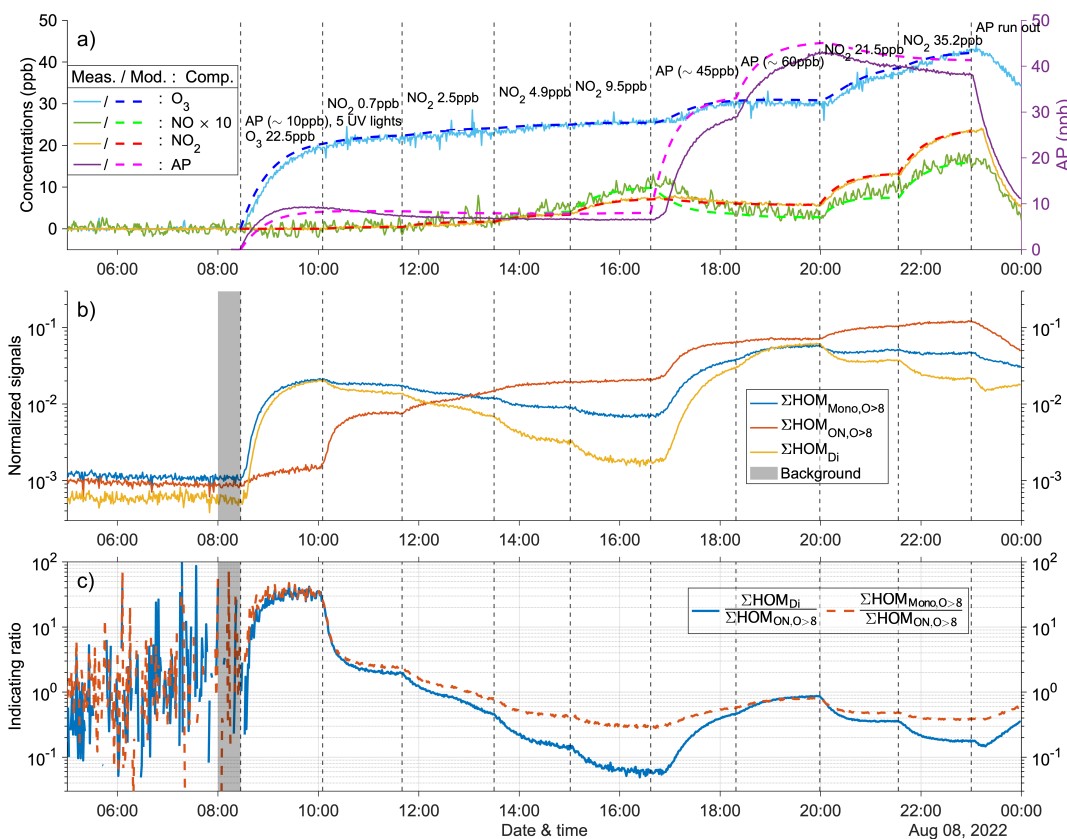

**Figure A8.** Time series of experiment no. 3 with 10 ppb $\alpha$-pinene and 22.5 ppb O$_3$ as initial inputs. 5 UV lights were on during all stages. Three subplots show the time series of different compounds: a) measured (abbreviated as Meas., in solid lines) and modeled (abbreviated as Mod., in dashed lines) concentrations of O$_3$, NO$_x$ (both shown by left y-axis; the NO concentration is multiplied by 10), and AP (i.e., $\alpha$-pinene, shown by right y-axis); b) normalized signals of $\sum\text{HOM}_{\text{Mono,O>8}}$ (sum of non-nitrate HOM monomers with more than 8 oxygen atoms), $\sum\text{HOM}_{\text{ON,O>8}}$ (sum of HOM organic nitrates with more than 8 oxygen atoms), and $\sum\text{HOM}_{\text{Di}}$ (sum of HOM dimers); c) IR1 ($\frac{\sum\text{HOM}_{\text{Di}}}{\sum\text{HOM}_{\text{ON,O>8}}}$) and IR2 ($\frac{\sum\text{HOM}_{\text{Mono,O>8}}}{\sum\text{HOM}_{\text{ON,O>8}}}$). The grey shaded area represents the time period selected for background subtraction before calculating the ratio. Dashed vertical lines indicate specific time points of operations, with the corresponding labels for each operation in the subplot a.

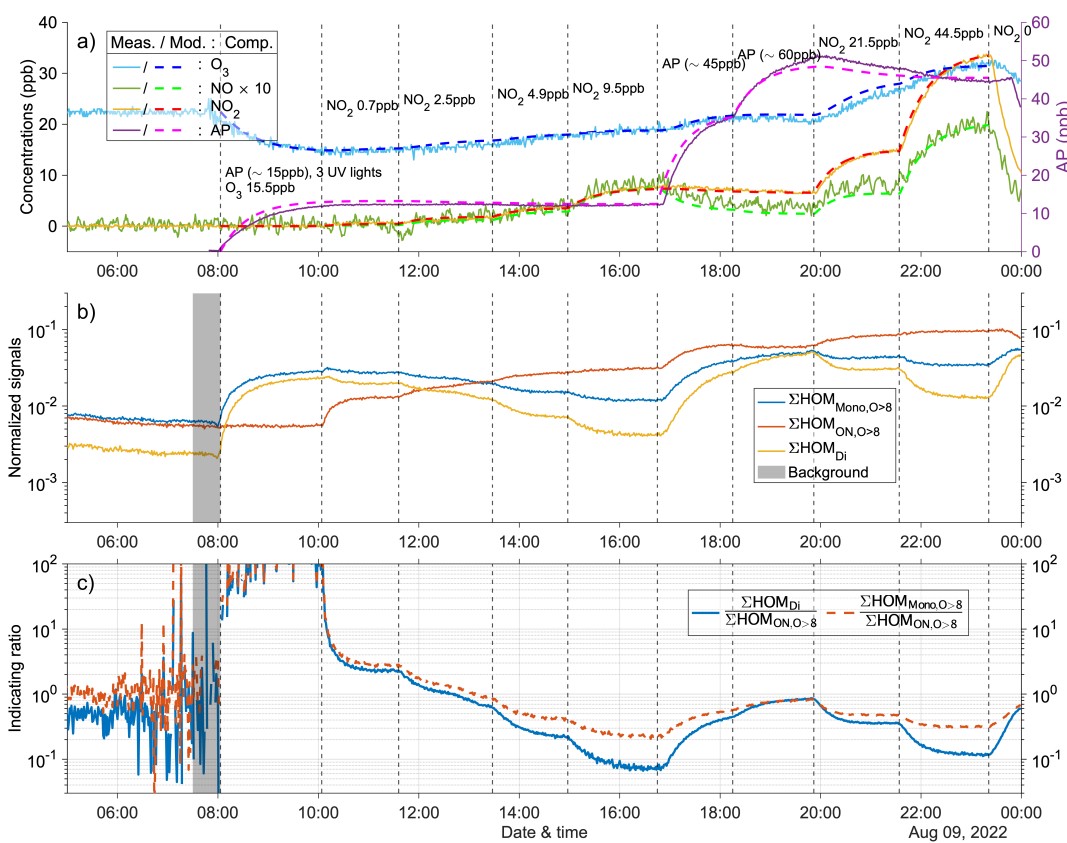

**Figure A9.** Time series of experiment no. 4 with 15 ppb $\alpha$-pinene and 15.5 ppb $O_3$ as initial inputs. 3 UV lights were on during all stages. Three subplots show the time series of different compounds: a) measured (abbreviated as Meas., in solid lines) and modeled (abbreviated as Mod., in dashed lines) concentrations of $O_3$, $NO_x$ (both shown by left y-axis; the NO concentration is multiplied by 10), and AP (i.e., $\alpha$-pinene, shown by right y-axis); b) normalized signals of $\sum HOM_{Mono,O>8}$ (sum of non-nitrate HOM monomers with more than 8 oxygen atoms), $\sum HOM_{ON,O>8}$ (sum of HOM organic nitrates with more than 8 oxygen atoms), and $\sum HOM_{Di}$ (sum of HOM dimers); c) IR1 ($\frac{\sum HOM_{Di}}{\sum HOM_{ON,O>8}}$) and IR2 ($\frac{\sum HOM_{Mono,O>8}}{\sum HOM_{ON,O>8}}$). The grey shaded area represents the time period selected for background subtraction before calculating the ratio. Dashed vertical lines indicate specific time points of operations, with the corresponding labels for each operation in the subplot a.



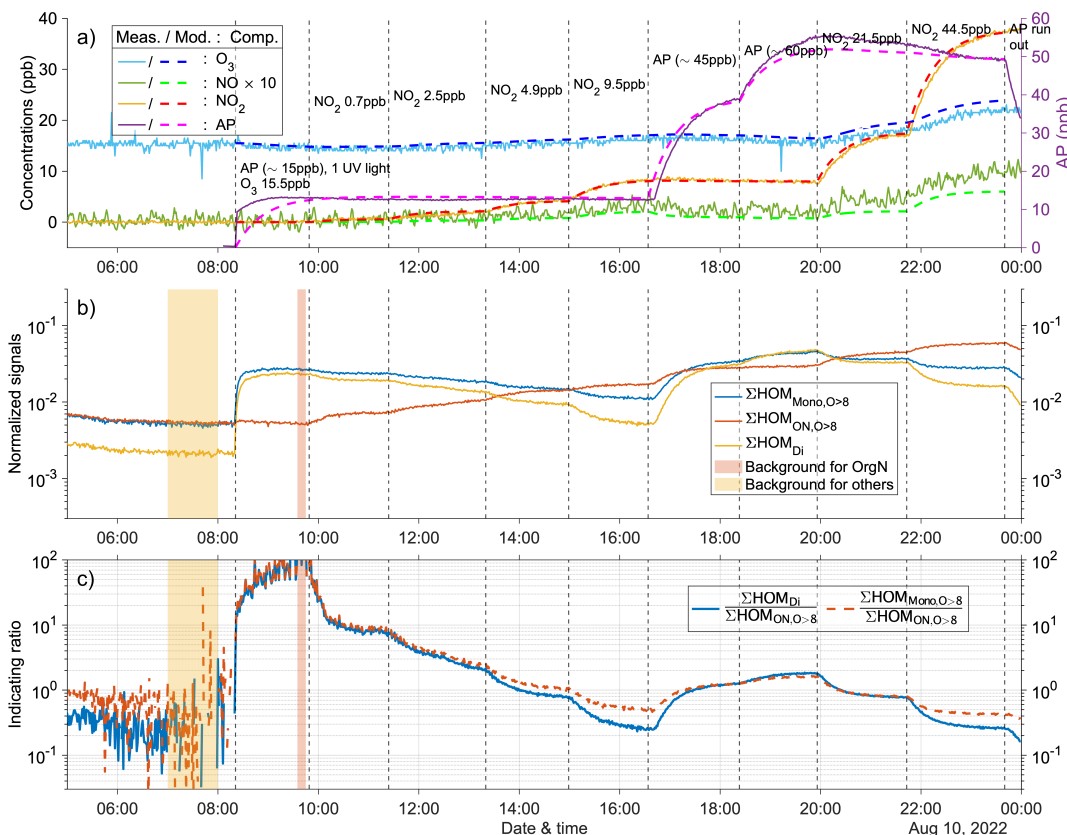

**Figure A10.** Time series of experiment no. 5 with 15 ppb $\alpha$-pinene and 15.5 ppb $O_3$ as initial inputs. 1 UV light was on during all stages. Three subplots show the time series of different compounds: a) measured (abbreviated as Meas., in solid lines) and modeled (abbreviated as Mod., in dashed lines) concentrations of $O_3$, $NO_x$ (both shown by left y-axis; the NO concentration is multiplied by 10), and AP (i.e., $\alpha$-pinene, shown by right y-axis); b) normalized signals of $\sum HOM_{Mono,O>8}$ (sum of non-nitrate HOM monomers with more than 8 oxygen atoms), $\sum HOM_{ON,O>8}$ (sum of HOM organic nitrates with more than 8 oxygen atoms), and $\sum HOM_{Di}$ (sum of HOM dimers); c) IR1 ($\frac{\sum HOM_{Di}}{\sum HOM_{ON,O>8}}$) and IR2 ($\frac{\sum HOM_{Mono,O>8}}{\sum HOM_{ON,O>8}}$). The yellow and red shaded area represent the time periods selected for background subtraction of dimers and organic nitrates, before calculating the ratio. Dashed vertical lines indicate specific time points of operations, with the corresponding labels for each operation in the subplot a.

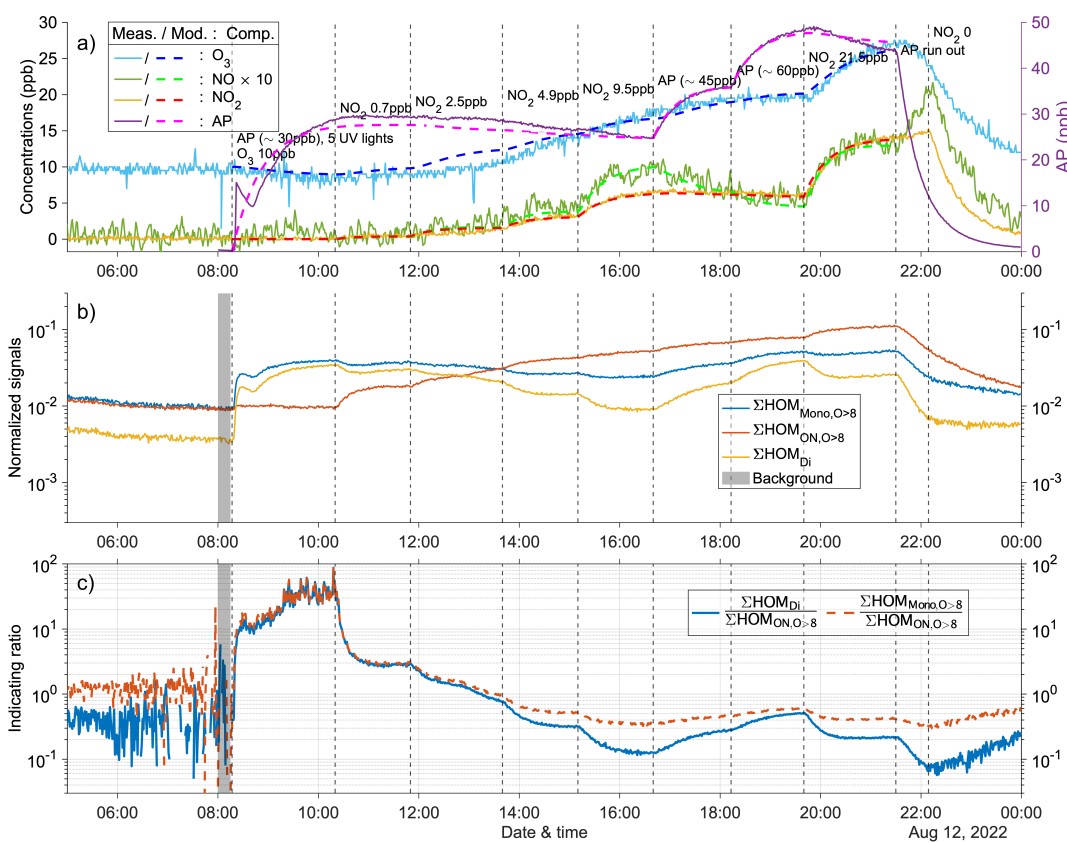

**Figure A11.** Time series of experiment no. 7 with 30 ppb $\alpha$-pinene and 10 ppb $O_3$ as initial inputs. 5 UV lights was on during all stages. Three subplots show the time series of different compounds: a) measured (abbreviated as Meas., in solid lines) and modeled (abbreviated as Mod., in dashed lines) concentrations of $O_3$, $NO_x$ (both shown by left y-axis; the NO concentration is multiplied by 10), and AP (i.e., $\alpha$-pinene, shown by right y-axis); b) normalized signals of $\sum HOM_{Mono,O>8}$ (sum of non-nitrate HOM monomers with more than 8 oxygen atoms), $\sum HOM_{ON,O>8}$ (sum of HOM organic nitrates with more than 8 oxygen atoms), and $\sum HOM_{Di}$ (sum of HOM dimers); c) IR1 ($\frac{\sum HOM_{Di}}{\sum HOM_{ON,O>8}}$) and IR2 ($\frac{\sum HOM_{Mono,O>8}}{\sum HOM_{ON,O>8}}$). The grey shaded area represents the time period selected for background subtraction before calculating the ratio. Dashed vertical lines indicate specific time points of operations, with the corresponding labels for each operation in the subplot a.





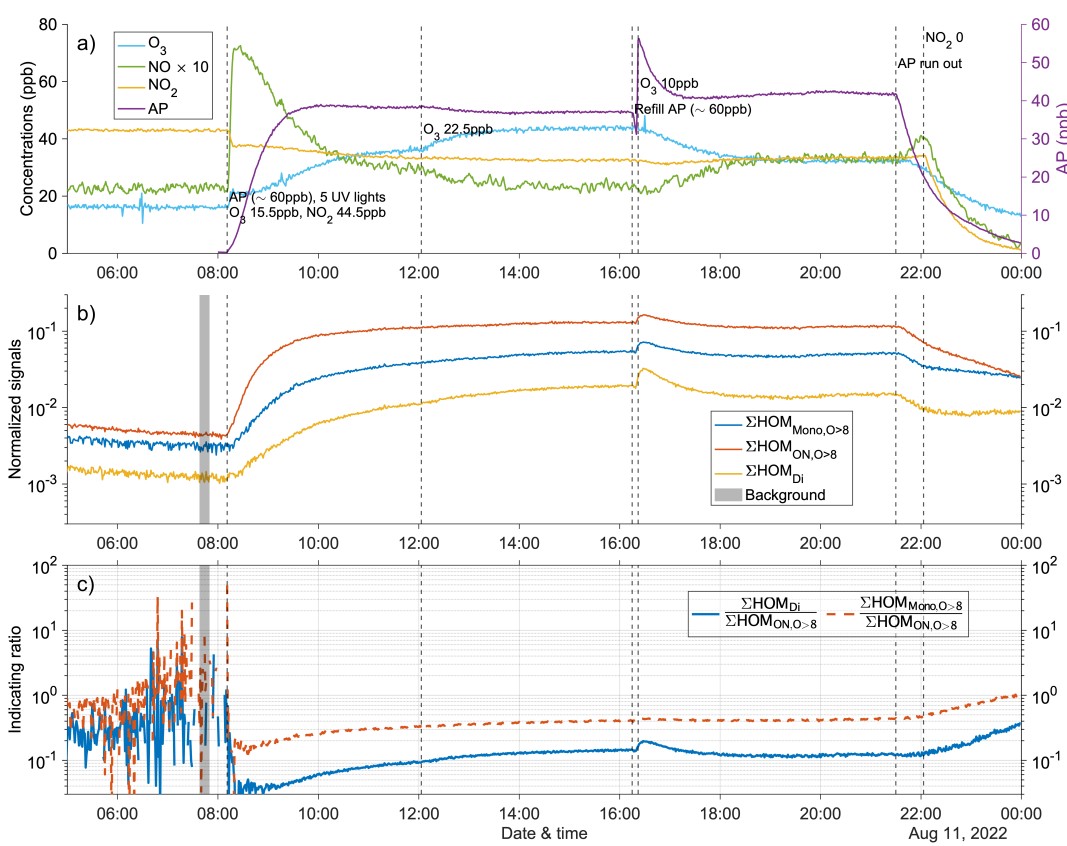

**Figure A12.** Time series of experiment no. 6, which was meant for collecting more data at the highest $NO_2$ input (at $\sim$ 44.5 ppb). 5 UV lights was on during all stages. Three subplots show the time series of different compounds: a) measured concentrations of $O_3$, $NO_x$ (both shown by left y-axis; the NO concentration is multiplied by 10), and AP (i.e., $\alpha$-pinene, shown by right y-axis); b) normalized signals of $\sum HOM_{Mono,O>8}$ (sum of non-nitrate HOM monomers with more than 8 oxygen atoms), $\sum HOM_{ON,O>8}$ (sum of HOM organic nitrates with more than 8 oxygen atoms), and $\sum HOM_{Di}$ (sum of HOM dimers); c) IR1 ($\frac{\sum HOM_{Di}}{\sum HOM_{ON,O>8}}$) and IR2 ($\frac{\sum HOM_{Mono,O>8}}{\sum HOM_{ON,O>8}}$). The grey shaded area represents the time period selected for background subtraction before calculating the ratio. Dashed vertical lines indicate specific time points of operations, with the corresponding labels for each operation in the subplot a.





**Figure A13.** Steady state IR2 ($\frac{\sum \mathrm{HOM_{Mono,O>8}}}{\sum \mathrm{HOM_{ON,O>8}}}$) of experiments from four days with 5 UV lights. X-axis is the multiplication of measured (steady state) $\alpha$-pinene and $O_3$ concentrations, while y-axis is the measured (steady state) $NO_x$. The scatter points (exp. no. 1 (experiment number 1): diamond; exp. no. 2: star; exp. no. 3: round; exp. no. 7: square) are colored by values of IR2 (abbreviated as R. in the figure), and are connected by curves (exp. no. 1: blue; exp. no. 2: orange; exp. no. 3: green; exp. no. 7: purple) showing the sequence (Seq.) of experimental stages. Panel a combines stages of all four days, and the rest three subplots respectively show the stages of experiments with different initial (Ini.) inputs (exp. no. 1 and 7 are in the same panel d due to same initial inputs). EKMA curves (isopleth of $O_3$ concentrations in ppb), simulated by the box model, are black solid lines, while dotted lines are corresponding ridge lines.





**Figure A14.** Steady state IR2 ($\frac{\sum \text{HOM}_{\text{Mono,O>8}}}{\sum \text{HOM}_{\text{ON,O>8}}}$) of experiments from three days with 5, 3, and 1 UV lights, respectively. X-axis is the multiplication of measured (steady state) $\alpha$-pinene and $O_3$ concentrations, while y-axis is the measured (steady state) $NO_x$. The scatter points (exp. no. 2 (experiment number 2): star; exp. no. 4: diamond; exp. no. 5: round) are colored by values of IR2 (abbreviated as R. in the figure), and are connected by curves (exp. no. 2: orange; exp. no. 4: blue; exp. no. 5: green) showing the sequence (Seq.) of experimental stages. Panel a combines stages of all three days with the same initial (Ini.) inputs, and the rest three subplots respectively show the stages of experiments with different amount of UV lights. EKMA curves by the box model are in black lines and the dotted lines are corresponding ridge lines.



**Table A1.** Zero-VOC experiment conditions. Experiment numbers (No.) and number of total stages are shown in the first two columns. Input information include the number and NO$_2$ photolysis rates ($J_{NO_2}$) of UV lights ($\lambda \approx 400$ nm) and concentrations of O$_3$, $\alpha$-pinene, and NO$_x$.

| Experiment *No.* | Number of stages | Input | | | |
|---|---|---|---|---|---|
| | | Number of lights | $J_{NO_2}$ (s$^{-1}$) | O$_3$ (ppb) | NO$_x$ range (ppb) |
| *Z1./Z2./Z3.* | 9/7/7 | 5 | $1.85 \times 10^{-3}$ | 10/15.5/22.5 | 0.7 - 44.5 |
| *Z4.* | 8 | 3 | $1.15 \times 10^{-3}$ | 15 | 0.7 - 44.5 |
| *Z5.* | 7 | 1 | $0.35 \times 10^{-3}$ | 15.5 | 0.7 - 44.5 |

The figures are shown....

**Table A2.** Chemical reactions and their reaction rate coefficients used for the box model. Note that RO$_2$ represent all kinds of peroxy radicals, thus there are huge uncertainties regarding reaction rates. This model is only meant for simulating concentrations of O$_3$ and its precursors.

| Chemical reactions | Reaction rate coefficients [a] |
|---|---|
| 1. $NO_2 + hv \xrightarrow{O_2} NO + O_3$ | $0.35/1.15/1.85 \times 10^{-3}$ |
| 2. $O_3 + NO \rightarrow NO_2 + O_2$ | $1.8 \times 10^{-14}$ |
| 3. $\alpha-pinene + O_3 \xrightarrow{O_2} RO_2 + OH$ | $8.7 \times 10^{-17}$ |
| 4. $\alpha-pinene + OH \xrightarrow{O_2} RO_2$ | $5.5 \times 10^{-11}$ |
| 5. $RO_2 + NO \rightarrow RO + NO_2$ | $1 \times 10^{-11}$ |
| 6. $RO_2 + RO_2 \rightarrow 2RO + O_2$ | $8 \times 10^{-13}$ |
| 7. $RO + O_2 \rightarrow HO_2$ | Instantaneous |
| 8. $NO + OH \rightarrow HNO_2$ | $4 \times 10^{-11}$ |
| 9. $NO + OH \xrightarrow{O_2} HO_2 + NO_2$ | $1.2 \times 10^{-13}$ |
| 10. $NO_2 + OH \rightarrow HNO_3$ | $4 \times 10^{-11}$ |
| 11. $HO_2 + NO \rightarrow OH + NO_2$ | $1 \times 10^{-11}$ |
| 12. $HO_2 + NO_2 \rightarrow HO_2NO_2$ | $3 \times 10^{-14}$ |
| 13. $HO_2 + HO_2/RO_2 \rightarrow peroxides$ | $2 \times 10^{-11}$ |

[a] Except the NO$_2$ photolysis rates (s$^{-1}$), all the other reaction rates (cm$^3 \cdot$s$^{-1}$) are adapted from NIST (National Institute of Standards and Technology) Chemical Kinetics Database (more information see https://kinetics.nist.gov/kinetics/index.jsp).





*Author contributions.* ME, JZ, and JYZ designed the study. JZ, JYZ, YL, and VM conducted the experiments. JYZ analyzed the data and developed the model. ME, JZ, and DR supported the data analysis.

*Competing interests.* Douglas Worsnop works for Aerodyne Research, Inc.

*Acknowledgements.* The authors thank Lauriane Quéléver for the calibration of the $NO_3$-CIMS.

*Financial support.* This work was supported by funding from Academy of Finland (grant no. 345982) and the Jane and Aatos Erkko Foun-
dation.

Open-access funding was provided by the Helsinki University Library.



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
