# Peer review of "On the potential use of highly oxygenated organic molecules (HOM) as indicators for ozone formation sensitivity"

_EGUsphere, 2023_

## Author Comment (AC1)

**On the potential use of highly oxygenated organic molecules (HOM) as indicators for ozone formation sensitivity**

Jiangyi Zhang[1], Jian Zhao[1], Yuanyuan Luo[1], Valter Mickwitz[1], Douglas Worsnop[1,2], and Mikael Ehn[1]

[1]Institute for Atmospheric and Earth System Research/Physics, Faculty of Science, University of Helsinki, Helsinki, 00014, Finland

[2]Aerodyne Research Inc., Billerica, Massachusetts, 01821, United States

**Correspondence:** Jian Zhao (jian.zhao@helsinki.fi) and Jiangyi Zhang (jiangyi.zhang@helsinki.fi)

**Response to Reviewer #1**

**General comments**

This work details on how HOM can function as an indicator for determining the sensitivity of $O_3$ formation. The authors have clearly communicated their approach, results, and the potential limitations of the study. In general, the manuscript is well-written and of good scientific quality. Therefore, I would recommend this manuscript for publication with the minor re-works and additions outlined below.

We thank the reviewer for taking the time to review our manuscript and for the positive and insightful comments. We will answer the specific comments point-by-point below. The reviewer's comments are in *blue*, and our answers are in *black* with updated content in **bold**.

**Specific comments**

**Comment #1:**

How the photolysis rate of $NO_2$ was determined? Maybe a brief discussion (used expressions) can be included in the manuscript.

**Response:**

The photolysis rates of $NO_2$ were determined by varying the $J_{NO_2}$ parameter in the model until the simulated $O_3$ and $NO_x$ values agreed with the observations in the zero-VOC experiments (Fig. 1 and A2–A5). The values for $J_{NO_2}$ could also be computed from the observed steady-state and input concentrations of $NO_x/O_3$ for each condition. We changed the text in the manuscript to more clearly reflect the method we used. We also added the a statement in the footnotes of Table A1 to describe

20     the expression from which $J_{NO_2}$ could be numerically derived:

**From steady-state (ss in subscript) balance of the $\mathbf{O_3}$ concentration, we can write the following expression:**

$$\mathbf{\frac{d[O_3]}{dt} = J_{NO_2}[NO_2]_{ss} + \frac{[O_3]_{input} - [O_3]_{ss}}{\tau} - k_{O_3,NO}[O_3]_{ss}[NO]_{ss} = 0}$$

**where $\tau = \frac{2000\ \mathbf{L}}{55\ \mathbf{L\ min^{-1}}}$ is the residence time. $O_3$ has $NO_2$ photolysis ($\mathbf{J_{NO_2}[NO_2]_{ss}}$) and its input ($\frac{[O_3]_{input}}{\tau}$) as sources, with reaction to NO ($\mathbf{-k_{O_3,NO}[O_3]_{ss}[NO]_{ss}}$) and flush-out ($\frac{-[O_3]_{ss}}{\tau}$) as sinks. We can solve the equation to get $NO_2$**

25 **photolysis rate:**

$$\mathbf{J_{NO_2} = \frac{k_{O_3,NO}[O_3]_{ss}[NO]_{ss} + \frac{[O_3]_{ss} - [O_3]_{input}}{\tau}}{[NO_2]_{ss}}}$$

**This expression can be used for each steady state to estimate $\mathbf{J_{NO_2}}$ in the corresponding experiment.**

**Comment #2:**

It was not clear from the discussion (Section 2.3) and Table A2, whether wall losses have been accounted for or not. A
30     discussion on these losses could be interesting to see the nature of the effect on indicating ratios.

**Response:**

Thanks for pointing this out, we should have mentioned that we considered $RO_2$ with a wall loss lifetime of 400 s (Peräkylä et al., 2020). We made corresponding modifications in Table A2. Wall loss remains a minor loss pathway for $RO_2$, though, as the lifetime with respect to bimolecular reactions tend to dominate. In contrast, for closed shell species used for the indicating
35     ratios, wall losses are the dominant loss term, but these were not included in the model. We now clarified the discussion in section 3.2 to make it more clear, including statements on the choice of parameters used to calculate the indicating ratios. The choice of including only the most oxygenated, i.e., least volatile, species was done specifically in order to have very similar wall loss rates for the species, which in turn meant that the exact wall loss rates were not of any great significance.

**Comment #3:**

40     There can be a discussion on why IR1 holds a better potential than IR2 for indicating $O_3$ formation sensitivity.

**Response:**

As we discussed in lines 257-260, "...both indicating ratios are promising as indicators of $O_3$ formation sensitivity. However, in all time series, IR1 exhibited more pronounced changes compared to IR2 as we shifted the $O_3$ formation regimes." This highlights that IR1 may hold better potential for indicating $O_3$ formation sensitivity in the well-controlled chamber systems
45     we investigated, since the nitrates are solely from $RO_2+NO$ and the dimers solely from $RO_2+RO_2$. But HOM monomers can be from both of these reactions. On the other hand, in the real atmosphere, e.g., some polluted urban areas, where $RO_2$ mainly reacts with NO instead of another $RO_2$, we may not observe HOM dimers at all. In this case, IR2 would be better than IR1,

as discussed in more detail in section 3.5. Following the reviewer's suggestion, we added more discussion: **This highlights that in the well-controlled chamber systems we investigated, IR1 may hold better potential for indicating $O_3$ formation sensitivity in the absence of other perturbing factors. It can be explained by the fact that the nitrates are solely from $RO_2+NO$ and the dimers solely from $RO_2+RO_2$. But HOM monomers can be from both of these reactions. On the other hand, in the real atmosphere, IR1 is expected to be much less robust, as discussed in more detail in section 3.5.**

**Comment #4:**

It is specified that this chamber study can estimate indicating ratios in determining $O_3$ formation sensitivity, both qualitatively as well as quantitatively. I suggest adding a table that shows the estimated and measured values of $O_3$ concentration as well as the indicating ratios, which will make it easier for the readers to refer to the values.

**Response:**

This is an important comment, and it seems that there is a misunderstanding towards our use of the term "quantitatively". We wanted to express that even the absolute values of the indicating ratios can determine either VOC- or $NO_x$-limited regimes in our experiments, and we already "quantitatively" gave the thresholds (i.e., IR1/IR2: <0.2/0.4, VOC-limited; >0.5/0.7, $NO_x$-limited). We now realize that it can be confusing and misleading to use "quantitatively", and we opted to remove this term completely from this context. Our point is already made clear by the fact that the absolute IR values alone were able to determine the sensitivity regime.

**Technical comments**

**Comment #5:**

Line 205: Despite showing a faster decay compared to $HOM_{ON,O\leq8}$, non-nitrate HOM monomers with fewer than 9 oxygen atoms ($HOM_{Mono,O\leq8}$) also showed overall slow decays (Fig. 3). – This sentence needs to be rewritten.

**Response:**

**Additionally, non-nitrate HOM monomers with fewer than 9 oxygen atoms ($HOM_{Mono,O\leq8}$) also showed an overall slow decay (Fig. 3).**

**Comment #6:**

After Table A1, the line 'The figures are shown....' should be deleted or completed.

**Response:**

Thanks for pointing out the redundant sentence, and it has been deleted.

**75  References**

Peräkylä, O., Riva, M., Heikkinen, L., Quéléver, L., Roldin, P., and Ehn, M.: Experimental investigation into the volatilities of highly oxygenated organic molecules (HOMs), Atmospheric Chemistry and Physics, 20, 649–669, https://doi.org/10.5194/acp-20-649-2020, 2020.

---

## Author Comment (AC2)

**On the potential use of highly oxygenated organic molecules (HOM) as indicators for ozone formation sensitivity**

Jiangyi Zhang[1], Jian Zhao[1], Yuanyuan Luo[1], Valter Mickwitz[1], Douglas Worsnop[1,2], and Mikael Ehn[1]

[1]Institute for Atmospheric and Earth System Research/Physics, Faculty of Science, University of Helsinki, Helsinki, 00014, Finland

[2]Aerodyne Research Inc., Billerica, Massachusetts, 01821, United States

**Correspondence:** Jian Zhao (jian.zhao@helsinki.fi) and Jiangyi Zhang (jiangyi.zhang@helsinki.fi)

**Response to Reviewer #2**

**General comments**

The authors conducted chamber experiments to evaluate whether HOM can function as a real-time indicator of the $O_3$ production regime. They classified different types of HOM, and showed that the $O_3$ sensitivity can be estimated from the composition

5 of HOM (nitrate-containing and non-nitrate compounds) in a single VOC system.

Overall, this manuscript is well written, and the results are useful to researchers working on the $O_3$ formation mechanism.

We thank the reviewer for taking the time to review our manuscript and for the positive and insightful comments. We will answer the specific comments point-by-point below. The reviewer's comments are in *blue*, and our answers are in *black* with

10 updated content in ***bold***.

**Specific comments**

**Comment #1:**

The authors used the ratios of non-nitrate HOM and nitrate-containing HOM as the indicating ratios. The reason for using HOM

15 (not oxygenated VOC) is not clear to me. In general, HOM compounds have lower volatility than most OVOC compounds, and thus, they have more uncertainty factors, such as higher aerosol formation (for both ambient air and chamber experiments) and wall deposition (for chamber experiments). In this study, I guess the authors measured only gaseous HOM by CIMS, and thus, the production yield of HOM cannot be fully measured. The volatilities of HOM compounds varied widely, and thus, the fractions of HOM in a particulate phase or deposited on walls are different among compounds. The application of the

20 OVOC/organic nitrate ratio as the indicating ratio seems more straightforward than the HOM ratio proposed in this manuscript. The reason for using HOM should be clearly stated.

(In a similar context, I do not understand why compounds with O<=8 are not included in this analysis. Compounds with O>8 have lower volatility, and thus, aerosol formation or wall deposition would be expected to interfere with the estimation of the HOM production yields).

**Response:**

We thank the reviewer for this important comment, as it clearly indicates that we did not make this critical topic clear enough. The low volatility of HOM is very critical when we consider the features of potential indicating ratios. As we discussed e.g. in the Abstract line 15, Introduction line 106, and Conclusion line 334, the fast formation while very short lifetime makes HOM special and can be potentially used to track the instantaneous chemical regime of $O_3$-related photochemistry in the atmosphere (i.e., as sort of a real-time indicator). However, if we use less oxygenated species in this study, e.g., HOM with O<8 in Figure 3, these species will slowly but continuously evaporate off the wall after the previous experiment(s), interfering with their background measurements for the next experiment. Differently, the more oxidized HOM species decay faster and reach a relatively stable and lower level of background after the experiments. This is why we ended up with using these compounds. A similar argument stands for the real atmosphere, where HOM species condense similarly to the particle surfaces as they did onto the walls. Whereas, OVOC may linger in the air for hours to days, undergo multiple steps of oxidation and cycles before they are removed from the atmosphere. In this way, HOM is actually simpler than OVOC, thus more straightforward as especially a real-time indicator.

We only measured the gas-phase HOM, and in purpose. As explained in the next Comment (i.e. #2), we planned to run the experiments at low concentrations to prevent SOA formation, thus no condensation on particles. In addition, we are more interested in the effects of NO on the relative distribution of HOM, and how that relates to $O_3$ formation, instead of accurate HOM yield estimation. Overall, as the most oxygenated HOM species we selected had roughly comparable loss rates in the chamber (Fig. 3), the indicating ratios bear fewer uncertainties. We made modifications in section 3.2 to answer this comment as well as the comment #2 from reviewer #1.

**Comment #2:**

Is there the aerosol formation in this experimental system? If large fractions of the produced HOM are in a particulate phase, then, the interpretation of the derived indicating ratio becomes complicated. The information on aerosol formation in this experimental system is useful to readers.

**Response:**

There was no significant aerosol formation in our system, and the main loss of HOM was to the walls. Even if there had been aerosol formation, the nearly non-volatile behavior of the chosen HOM compounds would have caused a similar loss rate for all of them, thus not impacting the indicating ratios. We did connect a Differential Mobility Particle Sizer (DMPS), measuring particles larger than 10 nm, to our chamber briefly at the highest $\alpha$-pinene injection (at 60 ppb) stage, and there was no sign

of particle formation. Also, if there was particle formation, the time series of HOM species may not be so steady at the end of each stage because of condensation (Zhao et al., 2023). We added a short discussion at the beginning of the section 3.3 to

55 clarify this point: **It is worth noting that our experiments did not result in notable particle formation, and condensation to walls was always the dominant loss term for HOM. If aerosol formation had been significant, as has been observed in our chamber at higher oxidation rates (Zhao et al., 2023), HOM would first increase due to fast formation, and then decrease due to condensation sink.**

**Comment #3:**

60 The estimated threshold values of the indicating ratios are the key findings of this study, whereas the derivation of the thresholds is not explained in the main text. The derivation of these threshold values (shown in Lines 307-309) should be clearly explained.

**Response:**

Our key finding is the viability of using HOM ratios as real-time indicators, but we want to highlight that the estimated threshold values are limited only to our experimental system ($\alpha$-pinene + $O_3$ + $NO_2$). We have now added a statement at the end of section

65 3.2 to make this more clear. The threshold values were not derived rigorously from a mathematical or chemical point of view. Instead, they were summarized from Figures 5, 6, A13, and A14, based on modeled EKMA curves and experimental points. The idea of giving those values is showing the possibility of using the absolute values of HOM ratios, but as also discussed in the response to reviewer #1, we have opted to avoid using the term "quantitatively" to avoid misinterpretation of these absolute numbers. We also changed the relevant sentence: **More specifically, based on modeled EKMA curves and steady state**

70 **HOM ratios (Fig. 5, 6, A13, and A14), regardless of...**

**Technical comments**

**Comment #4:**

Figure 5: "Measured" should be removed from the labels of the x and y axes, because the results of EKMA are also shown in this figure.

75 **Response:**

We have removed the "Measured" from labels (for Fig. 5–6 and A13–A14) and also did corresponding changes in the figure caption.

**References**

Zhao, J., Häkkinen, E., Graeffe, F., Krechmer, J. E., Canagaratna, M. R., Worsnop, D. R., Kangasluoma, J., and Ehn, M.: A combined gas- and particle-phase analysis of highly oxygenated organic molecules (HOMs) from $\alpha$-pinene ozonolysis, Atmospheric Chemistry and Physics, 23, 3707–3730, https://doi.org/10.5194/acp-23-3707-2023, 2023.

80